# Turmell-Meter: A Device for Estimating the Subtalar and Talocrural Axes of the Human Ankle Joint by Applying the Product of Exponentials Formula

**DOI:** 10.3390/bioengineering9050199

**Published:** 2022-05-04

**Authors:** Óscar Agudelo-Varela, Julio Vargas-Riaño, Ángel Valera

**Affiliations:** 1Facultad de Ciencias Básicas e Ingeniería, Universidad de los Llanos, Villavicencio 500002, Colombia; oscar.agudelo@unillanos.edu.co; 2Instituto Universitario de Automática e Informática Industrial (Instituto ai2), Universitat Politècnica de València, 46022 Valencia, Spain; giuprog@isa.upv.es

**Keywords:** human ankle model, product of exponentials formula, anthropometry, biomechanics, coordinate measuring machines, kinematics, pose estimation, position measurement, biomedical informatics

## Abstract

The human ankle is a complex joint, most commonly represented as the talocrural and subtalar axes. It is troublesome to take in vivo measurements of the ankle joint. There are no instruments for patients lying on flat surfaces; employed in outdoor or remote sites. We have developed a “Turmell-meter” to address these issues. It started with the study of ankle anatomy and anthropometry. We also use the product of exponentials’ formula to visualize the movements. We built a prototype using human proportions and statistics. For pose estimation, we used a trilateration method by applying tetrahedral geometry. We computed the axis direction by fitting circles in 3D, plotting the manifold and chart as an ankle joint model. We presented the results of simulations, a prototype comprising 45 parts, specifically designed draw-wire sensors, and electronics. Finally, we tested the device by capturing positions and fitting them into the bi-axial ankle model as a Riemannian manifold. The Turmell-meter is a hardware platform for human ankle joint axes estimation. The measurement accuracy and precision depend on the sensor quality; we address this issue by designing an electronics capture circuit, measuring the real measurement with a Vernier caliper. Then, we adjust the analog voltages and filter the 10-bit digital value. The Technology Readiness Level is 2. The proposed ankle joint model has the properties of a chart in a geometric manifold, and we provided the details.

## 1. Introduction

Taking in vivo measurements in the human ankle joint is troublesome because the ankle is a complex mechanism [1]. Deviations in the axis increase the pronation or supination moments, causing instability and enhancing injuries risk. In this work, we present a device intended for the study of the human ankle joint (HAJ). Modeling and measuring this lower limb joint is essential in physiology, biomechanics, and rehabilitation (also in humanoid robotic limb development).

Our primary aim is to develop a device for the two axes model estimation of the human ankle joint. Secondary objectives are: it must be non-invasive, compact, energy-efficient, and easy to set up and transport. It should also be compatible with laying positions, such as with the foot in the elevated position. To accomplish the objectives, we followed a plan, first by understanding the ankle movements. Then, we used statistics for dimensional determination. We also use a modern approach, such as the Product of Exponential (POE) formula. We then designed the structure based on embedded non-invasive distance sensors.

Our contribution to the ankle joint axis localization is the holistic development of a specific device. Draw-wire sensors measure distance, are composed of a wire wound around a drum, and are attached to a potentiometer and a spring. They are retractile with constant tension. For bias correction and gain calibration, we designed a capture system. We adjust the voltage to avoid the maximal value of the analog-to-digital conversion. We calibrated each sensor through direct measuring with a Vernier caliper. Then, we measured the voltage and adjusted the offset and gain by a calibration program in Processing (Software). Limitation measurements are by 10-bit analog-to-digital converters and digitally filtered in the acquisition board. Technology Readiness Level (TRL) is 2.

We highlight our approach over traditional methods because we apply the POE formula to the ankle kinematic model. Furthermore, we estimate the ankle axis localization by a geometric approach, solved algebraically. We computed it from the pseudo-inverse application. For the talocrural and subtalar axes estimation, we use circle fitting. As an alternative ankle joint representation, we propose a Riemannian chart. We have limited the scope to the human ankle joint (HAJ) model. There are applications in physical therapy and HAJ mobility diagnosis.

The state of the art in the ankle localization is detailed in [1,2,3,4,5,6,7,8,9,10,11,12,13,14].

There are different HAJ models in the literature; we focus on the two-axes approach. The approach is recommended by the International Society of Biomechanics (ISB) [15], anatomy and biomechanics books [3,16,17,18,19], and simulation software [20]. We found models of the ankle joints in several articles [14,21,22,23,24,25,26]. Contributions to the study of the ankle joint axes are in [2,8,9,27]. The most cited research about the subtalar axis are in [5,7,10,11,12,13]. A literature review of functional representations is in [4].

Draw-wire sensors (DWS) are distance measurement sensors, who use a wire coiled on a drum attached to a potentiometer and a spiral spring that are retractile at constant tension. Similar robotic applications are in [28,29,30], also in linear position tracking [31], and easy robot programming [32]. Inertial measurement units (IMU) were post-processed and complemented with other sensors [33,34,35,36,37]. We shall employ our device for the HAJ bi-axial measurements and for other models as well [38]. Biodex^TM^ and Humacnorm^TM^ are manufacturers of general kinetics machines.

We divide the materials and methods section into two subsections: the motion theory and the mechatronics design. In the first section, we study anatomy, statistics, proportions, and anthropometry to understand the functional HAJ movements and standard dimensions. Then we perform the HAJ simulation using the POE formula. Here, we do not include a deep study of infinitesimal kinematics. We intend to design a device for a healthy HAJ with no singularities with a continuous range of movement. We describe the trilateration method to find the platform pose. It is a geometrical method based on tetrahedrons; we avoid numerical solutions that depend on finite derivative terms. The tetrahedron is a well-defined 3D geometrical structure. Solving tetrahedron geometry is the expansion of planar trigonometry. Knowing the sides allows us to find the height of a tetrahedron. We attach the platform to the foot; the sensors are passive elements and do not support or add high tensile forces. We have selected the first seven sensor configurations 3-2-2 (seven sensors) instead of 3-3-3 (nine sensors) or 3-2-1 (five sensors) for hardware limitations, sensor redundancy, and symmetrical design (for both limb use).

The device’s mechatronics design and implementation are in the second subsection. We used Draw-wire sensors to measure the tetrahedron sides. These sensors have a constant tension because they comprise a drum attached to a spiral spring. We limit them to the maximal distance, and the precision depends on the potentiometer and electronics signal conditioning with a high common-mode rejection ratio (CMRR). The calibration process deals with accuracy and precision. First, we made rough adjustments to the acquisition system. Second, the software calibration process makes fine adjustments. Our proposed method avoids numerical errors because it uses geometric formulas. We validate the position through sensor redundancy. We conduct calibration and testing in a healthy patient and represent the HAJ movements as a manifold chart. The complementary source code was uploaded to [39].

## 2. Materials and Methods

This section is grouped in two main subsections, first the motion theory, and second the mechatronic system. For the first part, we show the simulation using anthropometric values and the POE formula. Using the plots, we estimate the DWS maximal length. Next, we present the device’s geometrical design and the trilateration method. Finally, we compute the axis position by circle fitting and modeling the ankle joint as a Riemannian manifold chart. In the second subsection, we describe the mechanics design and implementation, we used SolidWorks^®^ (2017–2018 Student Edition, Dassault Systèmes, Vélizy-Villacoublay, France), KiCad^©^(6.0.4, Jean-Pierre Charras and KiCad developers, CERN, Linux Foundation), and FreeCad^©^ (0.19, Jürgen Riegel, Werner Mayer, Yorik van Havre and others) StepUp tools addon.

### 2.1. Motion Theory

For the simulation with the POE formula, we adapt the data from [40], proportions from [41,42], and statistics from [43].

#### 2.1.1. References Assignation

Figure 1 presents the reference points and the mean distances taken from [40].

A, B, and C are the triangle’s vertices in a platform fixed to the foot, the K, L, and O distances from the most medial and lateral points from the black-filled to the white-filled marker. $M_{1}$ and $M_{2}$ define the talocrural (TC) axis. We show top-transverse and right-lateral views in Figure 2 with distances Q, W, and w. $N_{1}$ and $N_{2}$ determine the subtalar (ST) axis.

Table 1 enumerates the mean values of Figure 1 and Figure 2.

In Figure 3, we show the ST and TC axes from several viewpoints. The TC axis refers to the sagittal plane and the ST to the transverse plane.

#### 2.1.2. Anatomical and Geometrical Correspondence

We define the sagittal (lateral) plane as the X-Z plane (perpendicular to the *y*-axis). The coronal (frontal) plane is the Y-Z plane (*x*-axis is normal to it); the transverse (axial) plane is the X-Y plane (perpendicular to the *z*-axis). Figure 4, left, shows this corresponding references.

With this reference frame, we can define the TC axis orientation from a unitary vector in the z-direction. We first rotate it −80° around the *x*-axis; then we turn it −6° around the *z*-axis. A unitary vector in the *x*-axis direction defines the ST axis, rotating 41° about the *y*-axis, followed by a 23° rotation around the *z*-axis.

We show the fibula, tibia, talus, calcaneus 3D position, reference points, TC, and ST axes in Figure 4, right.

In this image, $A_{0}$, $B_{0}$, and $C_{0}$ are the vertices from the platform fixed to the foot, and $P_{M}$ is the triangle’s center. $S_{1}$, $S_{2}$, and $S_{3}$ are fixed to the shank relative to the origin point $P_{0}$. $M_{1}$ and $M_{2}$ define the TC axis; $N_{1}$ and $N_{2}$ correspond to the ST axis. We define $r_{1}$ and $r_{2}$ as the sagittal plane intersection with the TC and ST axes.

#### 2.1.3. Size and Dimensions

This first part help us to determine the HAJ axes direction and orientation for some cases. However, it is difficult to design a device that fits all humans, and we cannot make a device that fits 90 % percentiles; we intend to design a device scalable and adjustable in a defined population group. We also make an effort to design adjustable foot and shank attachments. To do so, we select the device dimensions using the proportions extracted from [41]. The heigh is H, and the proportions we use are: distance from the knee to the foot is 0.285H, the distance from the ankle to the foot is 0.039H and the foot widht is 0.055H and the foot length is 0.152H.

We select the origin of coordinates between the knee and the ankle, $d_{m}$ is the distance from $P_{M}$ to $P_{O}$. This distance is proportional to the body’s height H. To do so, we define $d_{m}$ as follows:(1)$$d_{m}=∥P_{0}-P_{M}∥=\left[\frac{0.285-0.039}{2}+0.039\right]\cdot H=0.162\cdot H.$$

For the sake of obtaining the prototype dimensions, we use statistics for a specific population. In [43], the mean height H of an adult male is 175 cm; by substituting this value into the equation, the knee-ankle distance is 28.35 cm. The distance $d_{p12}$ between points $r_{1}$ and $r_{2}$ about the TC and ST axes on the sagittal plane is:(2)$$d_{p12}=∥r_{1}-r_{2}∥=Q,$$
the projection of the most medial point (MMP) on the sagittal plane is:(3)$$P_{\mathrm{MMP}}=\left(x_{\mathrm{MMP}},0,z_{\mathrm{MMP}}\right),$$
and for the most lateral point is:(4)$$P_{\mathrm{MLP}}=\left(x_{\mathrm{MLP}},0,z_{\mathrm{MLP}}\right).$$

The point $M_{1p}$ is the projection of $M_{1}$ on the sagittal plane; we calculate it from the P and O values.
(5)$$M_{1p}=\left(x_{\mathrm{MMP}}-P,0,z_{\mathrm{MMP}}-O\right),$$
also, $M_{2p}$ is $M_{2}$; we estimate the projection from L and K through:(6)$$M_{2p}=\left(x_{\mathrm{MLP}}-L,0,z_{\mathrm{MLP}}-K\right).$$

Therefore, the segment $\bar{M_{2}M_{1}}$ has the sagittal projection $\bar{M_{2p}M_{1p}}$; it has the same proportional relation R=W/w in respect to $\bar{M_{2p}r_{1}}$, then:(7)$$\frac{M_{2}-M_{1}}{M_{2}-r_{1}}=\frac{W}{w}=R,$$
solving for $r_{1}$ gives the following:(8)$$r_{1}=M_{2}-\frac{M_{2}-M_{1}}{R}.$$

By knowing the distance Q projected in the sagittal plane and $r_{1}$, the angle 41° we calculate $r_{2}$ from:(9)$$r_{2}=Q\left[\cos\left(41^{\circ }\right),0,\sin\left(41^{\circ }\right)\right]+r_{1},$$

The distance from the origin $P_{O}$ to the plantar surface of the foot is $d_{m}$, we choose a circumscribed equilateral triangle with vertices $A_{0},B_{0},C_{0}$ as the platform base. The coordinates of $A_{0}$ are:(10)$$A_{0}=\left(r_{p},0,-d_{m}\right),$$
for $B_{0}$ are:(11)$$B_{0}=\left(r_{p}\cos60^{\circ },r_{p}\sin60^{\circ },-d_{m}\right),$$
and for $C_{0}$:(12)$$C_{0}=\left(r_{p}\cos-60^{\circ },r_{p}\sin-60^{\circ },-d_{m}\right),$$
where $r_{p}$ is proportional to H, then:(13)$$r_{p}=\frac{2}{3}\cdot H.$$

In summary, we estimate $P_{0}$, $r_{1}$, $r_{2}$; and the platform’s vertices $A_{0}$, $B_{0}$, and $C_{0}$. They are not arbitrarily selected, on the contrary, we employed anthropometry, statistics, and proportions.

#### 2.1.4. Product of Exponentials Formula

In this section, we employ the PoE formula. We follow the intuitive concept that inter-bone contact surfaces determine HAJ movements. Therefore, we represent these movements as a Special Euclidean group SE(3) in matrix form:(14)$$g=\left[\begin{array}{cc} R & {\hat{p}}_{T}\\ 0_{1\times 3} & 1 \end{array}\right],$$
where $R_{3\times 3}$ is the rotation matrix and ${\hat{p}}_{T}$ is the translation vector.

For the initial point $A_{0}$:(15)$$g_{A}\left(0\right)=\left[\begin{array}{cc} I_{3\times 3} & {\hat{A}}_{0}\\ 0_{1\times 3} & 1 \end{array}\right],$$
for $B_{0}$:(16)$$g_{B}\left(0\right)=\left[\begin{array}{cc} I_{3\times 3} & {\hat{B}}_{0}\\ 0_{1\times 3} & 1 \end{array}\right],$$
and for $C_{0}$
(17)$$g_{C}\left(0\right)=\left[\begin{array}{cc} I_{3\times 3} & {\hat{C}}_{0}\\ 0_{1\times 3} & 1 \end{array}\right]$$

We define ${\hat{\omega }}_{1}=\left(\omega_{x1},\omega_{y1},\omega_{z1}\right)$ as a unitary vector for the TC axis direction given by:(18)$${\hat{\omega }}_{1}=\frac{M_{2}-M_{1}}{∥M_{2}-M_{1}∥},$$
and a directed vector ${\hat{r}}_{1}$ from $P_{O}$ to $r_{1}$ is:(19)$${\hat{r}}_{1}=r_{1}-P_{O},$$
then, an orthogonal vector to ${\hat{r}}_{1}$ and ${\hat{\omega }}_{1}$ is:(20)$${\hat{\nu }}_{\theta_{1}r_{2_{z}}}=-{\hat{\omega }}_{1}\times {\hat{r}}_{1},$$
together, ${\hat{\omega }}_{1}$ and ${\hat{\nu }}_{\theta_{1}r_{2_{z}}}$ compound the six-dimensional vector ${\hat{\xi }}_{1}$:(21)$$\hat{\xi_{1}}=\left(\begin{array}{c} {\hat{v}}_{1}\\ {\hat{\omega }}_{1} \end{array}\right).$$

In the same way, there are correspondent vectors for the TC axis:(22)$${\hat{\omega }}_{2}=\frac{N_{2}-N_{1}}{∥N_{2}-N_{1}∥},$$
(23)$${\hat{r}}_{2}=r_{2}-P_{O},$$
(24)$${\hat{\nu }}_{2}=-{\hat{\omega }}_{2}\times {\hat{r}}_{2},$$
and:(25)$$\hat{\xi_{2}}=\left(\begin{array}{c} {\hat{v}}_{2}\\ {\hat{\omega }}_{2} \end{array}\right).$$

We compute R for each joint i=1,2 from the Rodrigues’ formula:(26)$$e^{\left(\Omega_{i}\theta_{i}\right)}=I_{3\times 3}+\Omega \sin\theta_{i}+\Omega^{2}\left(1-\cos\theta_{i}\right),$$
where Ω is the skew symmetric matrix:(27)$$\Omega =\left[\begin{array}{ccc} 0 & -\omega_{zi} & \omega_{yi}\\ \omega_{zi} & 0 & -\omega_{xi}\\ -\omega_{yi} & \omega_{xi} & 0 \end{array}\right].$$

The exponential formula is:(28)$$e^{\xi_{i}\theta_{i}}=\left[\begin{array}{cc} e^{\Omega_{i}\theta_{i}} & \tau_{i}\\ 0_{1\times 3} & 1 \end{array}\right],$$
and, $\tau_{i}$ is translation vector:(29)$$\tau_{i}=\left(I_{3\times 3}-e^{{\hat{\omega }}_{i}\theta_{i}}\right){\hat{\omega }}_{i}\times \hat{\nu }+{\hat{\omega }}_{i}{\hat{\omega }}_{i}^{T}{\hat{\nu }}_{i}\theta_{i}$$

Points A, B, and C have invariant relative positions, and there are two rotating joints; the PoE formula for A is:(30)$$g_{A}=e^{{\hat{\xi }}_{1}\theta_{1}}e^{{\hat{\xi }}_{2}\theta_{2}}g_{A}\left(0\right)=\left[\begin{array}{cc} R & {\hat{p}}_{A}\\ 0 & 1 \end{array}\right],$$
where ${\hat{p}}_{A}$ is the instantaneous position vector of A, the PoE for B:(31)$$g_{B}=e^{{\hat{\xi }}_{1}\theta_{1}}e^{{\hat{\xi }}_{2}\theta_{2}}g_{B}\left(0\right)=\left[\begin{array}{cc} R & {\hat{p}}_{B}\\ 0 & 1 \end{array}\right],$$
and the PoE for C is:(32)$$g_{C}=e^{{\hat{\xi }}_{1}\theta_{1}}e^{{\hat{\xi }}_{2}\theta_{2}}g_{C}\left(0\right)=\left[\begin{array}{cc} R & {\hat{p}}_{C}\\ 0 & 1 \end{array}\right],$$

$\theta_{1}$ is the TC rotation angle from the zero position, and $\theta_{2}$ is the ST rotation from the zero position. For the sake of clarity, we show the section of the ankle with the vectors ${\hat{r}}_{1}$, ${\hat{\omega }}_{1}$, ${\hat{\nu }}_{1}$ and ${\hat{r}}_{2}$; also the points A, B, C, and $P_{O}$ in Figure 5.

#### 2.1.5. Forward Kinematics

In this subsection, we show the simulation of the movements of the ankle by using the measurements and the PoE. The code is in SageMath Computer Algebraic System (CAS), which lets us manage symbolic notation, and interactive plotting in a Jupyter notebook. All source was uploaded to Git-Hub [39].

The simulation plot for the platform’s central point is in Figure 6a. We show the points $P_{O}$, $A_{0}$, $B_{0}$, $C_{0}$, $r_{1}$, $r_{2}$, and the surfaces representing each group of movements. The forward kinematics with $\theta_{1range}=\theta_{2range}=\left[-15^{\circ },15^{\circ }\right]$ and $\theta_{1}=\theta_{2}=10^{\circ }$ is in Figure 6b. For $\theta_{1range}=\theta_{2range}=\left[-10^{\circ },10^{\circ }\right]$ and $\theta_{1}=\theta_{2}=5^{\circ }$ is in Figure 6c.

Such a representation lets us compute the ankle joint ROM in all directions. Groups of A, B, C, and PM movements are smooth surfaces or geometric manifolds. They have two DOF, with a limited domain due to the axes ROM.

#### 2.1.6. Geometric Design and Trilateration Method

In the last section, we note that in a healthy ankle, the range of motion from three points in a platform attached to the foot, pertain to a surface without singularities. Moreover, we note that we can trace tetrahedrons from the reference base on the shank to the platform points. Tetrahedrons can be solved by knowing the triangle base, and the sides. We choose the complete tetrahedron with three distance sensors in the point A for symmetry conservation in the case of using the device in the left or right foot. We avoid the use of numerical methods such as the Newton–Raphson (NR), for reducing time of computation. Furthermore, we choose the symmetry and redundancy in the apexes B and C. We realize that by knowing the platform dimensions, two sensors and the apex A coordinates, we can define a plane rotated with respect to the base and solve other tetrahedrons corresponding to the B and C apexes. We also take a holistic approach, we knew that micro-controller systems often have two cores and eight or ten analog to digital converter channels. We used 7 channels, leaving three for temperature, battery level, and voltage input detection.

Finally, based on such considerations, we show a geometric design in the Figure 7a platform center, and in Figure 7b are the vertices.

By considering the distances between the origin and the vertices, we estimate the DWS maximal length in every module.
(33)$$l_{max}=\max\left[∥p_{A}\left(\theta_{1},\theta_{2}\right)-A∥+r_{m}\right]$$

Here, $l_{max}$ is the maximal possible length from the triangular inequality, $p_{A}$ is the positions group in $g_{A}$, $r_{m}$ is the module’s radius, and $A_{B}$ is the base point.

The main design requirement is the localization of three points attached to the foot. We estimate the actual position employing a DWS array in a tetrahedral structure to find the apex, which is a platform vertex. In Figure 8 we show the design structure.

$P_{O}$ and $P_{M}$ are the base and platform reference frames. The platform has known dimensions and the number of sensors is seven. First, we compute $A_{p}$ from three distances: $l_{A1}=∥A_{p}-A_{1}∥$, $l_{A2}=∥A_{p}-A_{2}∥$, and $l_{A3}=∥A_{p}-A_{3}∥$. Then, we compute $B_{p}$ and $B_{p}$ apexes after $A_{p}$ employing two DWS. We summarize the method in a flowchart; Figure 9.

#### 2.1.7. Finding the Apex in Tetrahedron A

In this section, we compute the tetrahedron $T_{A}$ with base ${▵}_{A}=\left[A_{1},A_{2},A_{3}\right]$ and apex $A_{p}$. Figure 10 shows the method we use.

In Figure 10 we see that triangles ${▵}_{132}=\left[A_{1},A_{3},A_{p132}\right]$ and ${▵}_{231}=\left[A_{2},A_{3},A_{p231}\right]$ are two sides of the tetrahedron $T_{A}$ developed on the base plane.

We compute the $A_{p132}$ and $A_{p231}$ orthogonal projection on each adjacent side of the module base triangle $▵\left[A_{1},A_{2},A_{3}\right]$ by tracing a circle centered on $A_{1}$ with radius $∥A_{p}-A_{2}∥$ and the circle centered on $A_{3}$ with radius $∥A_{p}-A_{3}∥$; resulting in $A_{p132}$ and $A_{p131}$ intersection points. In addition, the circle centered on $A_{2}$ with radius $∥A_{p}-A_{2}∥$ intersects the circle centered in $A_{3}$ at points $A_{p231}$ and $A_{p232}$. The segment from $A_{p132}$ to $A_{p131}$ intersects the points defined by $A_{p231}$ and $A_{p232}$ at $A_{\mathrm{pxy}}$. In the case of tetrahedron $T_{A}$, we determine $A_{\mathrm{pxy}}=\left(\mathrm{Apx},\mathrm{Apy},0\right)$ as $A_{p}$ projection on the base plane. It is easy to realize that the height of $T_{A}$ is the absolute value of the $A_{\mathrm{pz}}$ coordinate. Then, we can find the distance from $A_{\mathrm{pxy}}$ to $A_{3}$ as a triangle $▵\left[A\mathrm{pxy},A_{3},A_{p}\right]$ side; the other is $A_{\mathrm{pz}}$, and the hypotenuse is the distance $l_{A3}=∥A_{p}-A_{3}∥$, then, $A_{\mathrm{pz}}$ is:(34)$$A_{\mathrm{pz}}=\sqrt{l_{A3}^{2}-{\left(A_{\mathrm{pxy}}-A_{3}\right)}^{2}}$$

#### 2.1.8. Tetrahedrons B and C Apexes

In this subsection, we show that, by knowing $A_{p}$, the point $B_{p}$ needs two sensors to be found. To determine the result of the tetrahedron $T_{B}$, we consider the base of a triangle $▵\left[B_{1},B_{3},A_{p}\right]$ in Figure 11a.

We compute the angle α from the XY plane to a normal vector ${\hat{n}}_{ApB}$:(35)$${\hat{n}}_{ApB}=\frac{\left(B_{3}-Ap\right)\times \left(B_{3}-Ap\right)}{∥\left(B_{3}-Ap\right)\times \left(B_{3}-Ap\right)∥},$$
and, the angle α is:(36)$$\alpha =acos({\hat{n}}_{ApB}\cdot {\hat{n}}_{z}),$$
where ${\hat{n}}_{z}$ is the unitary vector normal to the XY plane.

The tetrahedron sides are the lengths lB1=∥Bp−B1∥, lB3=∥Bp−B3∥, and $d_{ApBp}=∥Bp-Ap∥$. The rotation axis is in the direction B1−B3. The Bpr is Bps rotated α in angle about this axis. In Figure 11b, we show how to find the Bpr apex, similarly to that of a tetrahedron $T_{A}$. Finally, when Bpr is found, the contrary rotation about the axis B1−B3 gives the Bps.

There are two possible apex values: Bps1 over, and Bps2 below of the XY plane. We show the Bpr apex below the XY plane in Figure 12.

We use the same method to solve the $T_{C}$ apex. For the correct apex selection, the condition when the side of the platform distance $d_{CpBp}$ is:(37)$$d_{CpBp}=∥Bps-Cps∥.$$

#### 2.1.9. Procedure for Found Platform Positions

We must fix the shank and the foot to the base and platform. Then we mark the MMP and the MLP. To do so, we design a detachable reference point from module A. Initially, we attach the foot and the shank to the device, and then we mark and record the MMP and MLP; Figure 13 shows the detailed view.

We compute the platform position from the seven sensor lengths. The main steps for capture data series are:1.Capture the initial position at horizontal relative position from dr=IMU2−IMU1 readings;2.Compute jerk $jrk=|dr_{i}-dr_{i-1}|$;3.Move the foot continuously until jerk crosses zero again.

First, we capture the sensor lengths by activating a button in the computer software. Every time, we compute the absolute difference from IMU2 to the IMU1 readings. If the differences are constants, then there is no platform and base relative movement. We compute the jerk by relative acceleration differentiation. The data capturing process ends when the acceleration change crosses zero. Jerk changes activate the capture of IMU data.

The symbolic equations to find $A_{p}$, $B_{p}$, and $C_{p}$ from the captured data, were found by the SageMath CAS. By using the prototype dimensions and the sensor lengths, we compute the platform’s position and orientation. Here, the origin is from the initial DWS lengths $l_{Mi0}$, where M is the module A, B, or C; and *i* is the sensor number i=1,2,3.

After MLP and MMP registering, we attach the apex of module A to the platform, define the sagittal plane perpendicular to the ABC base plane, and intersect point A. By implementing the trilateration method mentioned before, we compute the points $A_{0}$, $B_{0}$, and $C_{0}$.

Figure 14a illustrate the point positions with the device in the initial portable configuration. The apexes’ computation are in Figure 14b–d.

#### 2.1.10. Computing the Axis Position and Direction

From the anthropometric values [40], we put a mean model in the Turmell-Meter. The TC axis will be defined by $M_{1_{0}}$ and $M_{2_{0}}$. The sagittal plane intersection with the $\bar{M_{1}M_{2}}$ segment is $r_{1}$. For example, the TC axis approximation is computed from most lateral point (MLP), the most medial point (MMP) and L, K, P, and O:(38)$$M_{1_{0}}=MLP-\left[L,0,K\right],$$
and:(39)$$M_{2_{0}}=MMP-\left[-P,0,O\right],$$
from these values, we solve for $r_{1}$ from the plane y=0 intersection with the line $L_{TC}$:(40)$$L_{TC}=V-M1=\rho \frac{\left(M_{2}-M_{1}\right)}{∥M_{2}-M_{1}∥},$$
where V is a point pertaining to $L_{TC}$.

The ST axis sagittal intersection $r_{2}$ initial point is:(41)$$r_{2}=r_{1}+\frac{\sqrt{2}}{2}\left[Q,0,Q\right].$$

Here, $r_{1}$ and $r_{2}$ are reference values computed from the previously mentioned anthropometric mean values. Such initial points are for reference, comparison and validation of the trilateration and regression method. The tracked trajectory data set is processed offline. We use the least squares normal vector to the plane, this direction is similar to the circle approximation. From here, we compute the TC axis first, and then the ST axis. To do so, we compute the TC axis position by employing dorsiflexion and plantarflexion, because the TC axis is the most dominant in such movements. The method used is circle fitting in a plane containing the trajectory points. A further model refinement can be made with optimization, and machine learning methods, such as gradient descent and the symbolic product of exponential formula.

First, we found the TC axis orientation $\omega_{1}$ by registering several trajectories. For each trajectory, we have a list of data points P=[x,y,z], which pertain to a plane:(42)ax+by+cz+d=0,
where *a*, *b*, and *c* are the components of a direction vector perpendicular to a plane containing the points. Solving out for *z*, we have the system:(43)$$\left[\begin{array}{ccc} x_{0} & y_{0} & 1\\ x_{1} & y_{1} & 1\\ \vdots & \vdots & \vdots \\ x_{n-1} & y_{n-1} & 1 \end{array}\right]\left[\begin{array}{c} a\\ b\\ d \end{array}\right]=-c\left[\begin{array}{c} z_{0}\\ z_{1}\\ \cdots \\ z_{n-1} \end{array}\right]$$
which has the form:(44)Ax=B
there are more equations than unknowns. From linear algebra and least squares we knew that the pseudo inverse is $A^{+}={\left(A^{T}A\right)}^{-1}A^{T}$, then a normal vector is:(45)$$\left[\begin{array}{c} a\\ b\\ d \end{array}\right]={\left(A^{T}A\right)}^{-1}A^{T}B$$

Now, we compute *c* by replacing *a*, *b*, *d* in the plane equation, and finally we get $\hat{n}={\left[a,b,c\right]}^{T}$. We found the angle between the normal plane and the X-Y plane, after knowing the normal vector by applying the Rodrigues’ formula, $\hat{v}=\hat{n}\times \hat{k}$, with $\hat{k}={\left[0,0,1\right]}^{T}$
(46)$$P_{r}=P\cos\left(\theta \right)+\left(\hat{v}\times P\right)\sin\left(\theta \right)+\hat{v}\left(\hat{v}\cdot P\right)\left(1-cos\theta \right).$$
where $\theta =\arccos\left(\frac{\hat{n}\cdot \hat{k}}{∥\hat{n}∥}\right)$.

After this, we estimate the plane, and rotate all the data points onto the X-Y plane. We search for a circle in the X-Y plane, and rearrange the equation for least squares estimation by using a variable substitution.
(47)$$\begin{array}{ccc} {\left(x-x_{c}\right)}^{2}+{\left(y-y_{c}\right)}^{2} & = & r^{2}\\ \left(2x_{c}\right)x+\left(2y_{c}\right)y+\left(r^{2}-x_{c}^{2}-y_{c}^{2}\right) & = & x^{2}+y^{2}\\ c_{0}x+c_{1}y+c_{2} & = & x^{2}+y^{2} \end{array}$$
where $c={\left[c_{0},c_{1},c_{2}\right]}^{T}$ with $c_{0}=2x_{c}$, $c_{1}=2y_{c}$, and $c_{2}=r^{2}-x_{c}^{2}-y_{c}^{2}$.

By taking the rotated points, $P_{r}$ we have a linear system:(48)$$\left[\begin{array}{ccc} x_{0} & y_{0} & 1\\ x_{1} & y_{1} & 1\\ \vdots & \vdots & \vdots \\ x_{n-1} & y_{n-1} & 1 \end{array}\right]\left[\begin{array}{c} c_{0}\\ c_{1}\\ c_{2} \end{array}\right]=\left[\begin{array}{c} x_{0}^{2}+y_{0}^{2}\\ x_{1}^{2}+y_{1}^{2}\\ \vdots \\ x_{n-1}^{2}+y_{n-1}^{2} \end{array}\right].$$
that has the form:(49)Ac=b

In this system, we have more equations than unknowns, then, we search for the c values that minimize the squared difference ${∥b-Ac∥}^{2}$.
(50)$$\underset{c\in R^{3}}{\arg\min}{∥b-Ac∥}^{2}.$$

We found the center point $C_{p}=\left[x_{c},y_{c}\right]$ and radius *r* by solving:(51)$$\begin{array}{ccc} 2x_{c} & = & c_{0}\\ 2y_{c} & = & c_{1}\\ r^{2}-x_{c}^{2}-y_{c}^{2} & = & c_{2} \end{array}.$$

Finally, we apply a rotation to the center in respect to the original plane. This point pertains to the TC axis. For each trajectory A, B, C, we get three planes, and three centers, the TC line direction is parallel to the planes’ normal vectors. The information is complete by determining the plane orientation.

The ST axis estimation is similar, but employs trajectories from inversion and eversion movements.

This is a basic estimation; by conducting optimization on the product of the exponential formula, we enhance the accuracy of the axis position estimation.

#### 2.1.11. Ankle Joint Movements as a Manifold

In this subsection, we explain how the centers $r_{1}$, $r_{2}$ and directions $\omega_{1}$, $\omega_{2}$ define a manifold representing the HAJ movements. The circle center points calculated pertain to the TC and ST axes; they are the initial data to fit the product of the exponential formula. In Figure 15a, we show the complete platform’s center point manifold. It is topologically similar to a torus.

A manifold chart represents the range of motion limits, we show an example of the geodesic as a trajectory on the manifold in Figure 15b; this explains how to map ankle coordinates, and a straight trajectory with initial velocity and no external force action. We have the data necessary for the line intersection with the sagittal plane, the center points, and the direction gives a line:(52)$${\hat{p}}_{l}={\hat{l}}_{0}+\hat{l}d,$$
where ${\hat{p}}_{l}$ is the parametric line, $\hat{l}$ is a parallel vector to it, ${\hat{l}}_{0}$ is a known vector in such line, and d∈R, replacing the parametric equation in the plane equation:(53)$$\left({\hat{p}}_{l}-{\hat{p}}_{0}\right)\cdot \left({\hat{n}}_{p}\right)=0,$$
where ${\hat{p}}_{0}$ is a known vector in the plane, and ${\hat{n}}_{p}$ is the plane’s normal vector, solving for *d*, gives:(54)$$d=\frac{\left({\hat{p}}_{0}-{\hat{l}}_{0}\right)\cdot {\hat{n}}_{p}}{\hat{l}\cdot {\hat{n}}_{p}},$$
and replacing in the TC axis line equation:(55)$$r_{1}=c_{1}+\omega_{1}d,$$
where $r_{1}$ is the TC axis intersection with the sagittal plane. The point $c_{1}$ is the center, and the axis direction $\omega_{1}$, both were found by circle fitting. Furthermore, packing in six dimensional Plücker line coordinates, we have:(56)$${\hat{m}}_{1}=r_{1}\times \omega_{1},$$
and the $l_{1}$ six dimensional vector is:(57)$$l_{1}=\left[\omega_{1_{x}}:\omega_{1_{y}}:\omega_{1_{z}}:m_{1_{x}}:m_{1_{y}}:m_{1_{z}}\right].$$

We include those data for the PoE formula simulation and the manifold representation.

### 2.2. Mechatronic System Design

In this section, we design DWS to measure the lengths of the tetrahedron sides; they are arranged as structural parts. Their maximal length estimation is from the forward kinematics simulation. We design the shank attachment from the dimensions, proportions, and statistical data.

#### 2.2.1. Draw-Wire Sensor

We use flat springs. They are not exposed to a high load against gravity, and are in two or three concurrent groups. In Figure 16, we depict the design, composed of three 3D printed parts, potentiometer, flat spring, bolts, and nuts.

A two-coil winch drives the potentiometer; a flat spring retracts a wire attached to the winch. When we pull the wire, the spring retracts it. The value of each turn is from the nominal value of the potentiometer, $R_{n}=2.2\;k\Omega$, divided into ten turns, that is 220Ω per turn. The diameter is D=3.8cm, the spring could be compressed in four turns. The maximal length is as follows:(58)$$l_{max}=4\cdot D\cdot \pi$$

Which is 47.75 cm approximately, this value is greater than $l_{max}$ for all groups of movements.

#### 2.2.2. Mechanical Parts

The attachment on the calf has a size according to the simulation. We use the mesh model of a leg to guide the shape of the calf support, as in Figure 17a. We also scale and divide this structure into seven parts for 3D printing. An aluminum tube is the support structure, as in Figure 17b, and a neoprene band attaches the shank to the support with Velcro fabric.

All the DWS modules are in a plate, the A module has three DWS, B and C modules has two DWS, as in Figure 18a. The design of the foot attachment is from standard measurements to adjust the foot’s length and width, as in Figure 18b.

#### 2.2.3. Electronics

Two operational amplifiers in instrumentation configuration are the base block of the acquisition system, as Figure 19 shows. We employ the KiCad software for the circuit design.

The voltage gain in the instrumentation amplifier is:(59)$$A_{v}=\frac{v_{o}}{v_{i}}=\left[1=\frac{R_{2}}{R_{1}}+\frac{2R_{2}}{R_{1}}\right],$$

By selecting $R_{2}=100\;k\Omega$, $R_{1}=1\;k\Omega$, and $R_{G}=5\;k\Omega$, the voltage gain is 141. With 34 mV as voltage input, we get:(60)$$v_{o}=A_{v}\cdot v_{i}=4.794\;V$$

The final acquisition circuit has seven instrumentation amplifiers, with bias and gain trimmers for calibration. We design the printed board circuit as an ™Arduino Mega 2560 Shield, and assemble the components to the board by throw-hole soldering. We feed the circuits with a power system with two 18650 Li-Ion batteries in series, a backup pack, a Battery Management System (BMS); a 5V buck and a 12V boost converters. The Figure 20 shows the schematics. Finally, we add connectors for the MPU, OLED, and Bluetooth modules.

#### 2.2.4. Electronics Casing

We export the KiCad printed circuit design to FreeCAD StepUp to design the case containing all the components, focusing on a compact configuration design. The two main electronic components are the Arduino Mega 2560 and the Orange Pi One single board computer. We place the components, such as the Dual Pole Dual Throw (DPDT) toggle switches, symmetrically on the box sides. Figure 21 shows the main sides and the final assembly of the electronics case.

Each box has attached components to optimize the space. We test every component, and then install the support structure.

#### 2.2.5. Final Mechanical Assembly

The prototype consist of 45 3D printed parts, the union of main components is by an 8 mm steel threaded rod. The sub-assemblies uses M3 bolts and nuts. Figure 22 shows the assembly CAD.

#### 2.2.6. Calibration and Validation Software

Calibration is with the Arduino board connected to the PC, running a calibration program in processing. The basic program reads the IMU measurements and captures readings from the draw-wire sensors through the ADC inputs. The raw data are integer values with signs 2 bytes wide, the two 1-byte registers converted to 2-byte integers. An exponentially weighted moving average (EWMA) algorithm filters the raw signals and sends them to the PC via a serial port. The lengths computed are from the initial values plus the scaled sensor inputs with:(61)$$l_{iMj}=d_{iMj}+\frac{m_{iMj}}{s_{iMj}},$$
here, $l_{iMj}$ is the length in cm from the *i* wire to the *j* module, $d_{iMj}$ is the initial distance, $m_{iMj}$ is the measured digital value, and $s_{iM}$ is the scale factor in digital units per cm.

We present a rendered image with a scaled 175 cm model in Figure 23.

## 3. Results

We organize this section as follows: first, we show the simulation; second, the final prototype; third, the trilateration and axis orientation; and finally, an ankle manifold representation.

### 3.1. Simulation Results

In this subsection, we use different values from Table 1 to estimate the work-space and range of motion. First, we show the variation of mean value results, and second the platform position simulation by changing the range of movement and angles.

#### Changing Statistical Mean Values

Figure 24a shows the complete manifold, taking into account the intervals $\theta_{1},\theta_{2}\in \left[-180^{\circ },180^{\circ }\right)$. It also shows the platform’s initial position, the TC axis reference, the initial ST reference, the initial orientation, and a parametric trajectory with equal angle rate variation. In Figure 24b is the attaching point A simulation; Figure 24c,d depicts the simulations of B and C, respectively.

In Figure 25a we show the platform’ central point simulation with variations of 10% below the statistical mean values; Figure 25b shows the simulation changing 10% over the statistical mean values; Figure 26a is the attaching point A simulation adding the 10% mean values; and Figure 26b subtracts 10% of the mean values. Figure 27a,b are the results for the platform attaching point B. We show the results for the attaching point C in Figure 28a,b.

Finally, by changing the range of maximum and minimum angles, an example of the interactive simulation is in Figure 29a,b. We capture the view of the sliders and also show the simulation rendering result.

### 3.2. Final Prototype

In this section, we describe the results of the TM design, which are the assembled device and calibration. We try several designs and finally the CAD model is in [44]. First, we show images of the connected electronics parts. Second, we assemble the structure and perform calibrations. Third, we probe the device in a healthy patient to validate the prototype adaptability. We print the structural parts using ABS and the draw-wire sensor using PLA; PETG is in the supports and the case.

#### 3.2.1. Printed and Connected Electronics

We place the electronics in each side. In Figure 30, the connections and box sides and charge of the batteries.

#### 3.2.2. Printed and Assembled Structure

We assemble all structural components carefully, putting them together with stainless-steel threaded rods; then we place the draw-wire sensors, the acquisition board, connections, and final structure for calibration. Figure 31 shows the assembly.

#### 3.2.3. Calibration Results

We calibrate the system by using a personal computer. The resulting calibration, and measures of the lengths, are in Figure 32. The lecture is at the initial position, then we compare with the SolidWorks^®^ model measurements and the Vernier caliper real measurements for each DWS. The Table 2 shows the calibration results.

Figure 33a shows the length with a SolidWorks^®^ Measurement tool for module A, sensor 1; the lecture for sensor 2 is in Figure 33b. In Figure 33c, is the sensor 3 length. Table 3 shows the error measured in the real prototype and in SolidWorks^®^.

### 3.3. Trilateration Results

In this section, we use the measurements from the sensors to compute trilateration, then we compare them with the simulation results. The foot and shank fit in the adjustable platform and support structure, respectively, as is shown in the initial procedure in Figure 13. By introducing the DWS lengths to the virtual model, we compute the A, B and C coordinates in four consecutive positions. In the Table 4 are the seven sensors lenghts, and in the Table 5, we show the A, B and C coordinates for the four positions. The resulting figures for the first two positions are in Figure 34a–b, and for the latest two positions in Figure 35a,b. We show the base triangles, the points, the sensors, the platform and the circles on the base.

### 3.4. TC Axis Circle Fitting

The results of circle fitting for trajectories A, B, and C are in Table 6, corresponding to ankle joint plantar/dorsiflexion movements. We show the circle fitting for trajectories A, B, C, and PM in the Figure 36a–d.

### 3.5. ST Axis Circle Fitting

The results of ST circle fitting for trajectories A, B, C, and PM are in Table 7, corresponding to ankle joint inversion movements. We show the circle fitting for trajectories A, B, C, and PM in the Figure 37a–d.

### 3.6. Ankle Manifold Representation

In this section, we show the results in the software SageMath Manifolds. We load the model and visualize it as a manifold, we show the axis and the sagittal plane intersection. With the model parameters loaded, $r_{1}$, $r_{2}$, $\omega_{1}$, $\omega_{2}$, and the origin established in the center of the base modules. We apply the equation:(62)$${\hat{r}}_{1}={\bar{c}}_{0}+{\hat{n}}_{p}\cdot d$$
where ${\bar{c}}_{0}$ is the median center computed from trajectories A, B, and C center fitting, and ${\hat{n}}_{p}$ is the median planes’ normal vectors containing the circles. Table 8 shows values for the TC axis in the PM chart. In Table 9, we show the Plucker coordinates for the TC and ST axes.

Finally, Figure 38a shows the ankle manifold, and Figure 38b, the chart representing the range of movement and angle coordinates.

## 4. Discussion

In this work, we addressed the human ankle joint model from an alternative approach. We used statistical measurements for the development of a new device, specially designed to capture the human ankle joint movements. In animal joints, it is difficult to place encoders and linear sensors to measure the range of movement of complex joints in each internal living tissue reference frame. The product of exponential formulas uses only two frames, and it is useful in this case. Furthermore, in our work, we used a trilateration method for finding the device’s platform position, which is an analytic method. Therefore we avoid numerical approximations that can diverge and reduce rounding errors. We proposed the ankle joint model as a Riemannian manifold. We can define a chart as a subset of such a manifold with angle coordinates for measuring the range of movement. Our presented device is lightweight, non-invasive, and can be used in remote places, on beds, or on the floor. By characterizing the ankle parameters, we can conduct symmetry studies by correlating the left and right ankle joints. We can enhance the device configuration in future versions by replacing the draw-wire sensors used from potentiometers to digital encoders connected by a CAN bus, reducing wiring, space, weight, and energy consumption. We will use the model for the synthesis and reconfiguration of an ankle parallel rehabilitation robot, programmed by symmetrical movements at the opposite ankle. By employing the axis location and the screw theory, forces, and torques, we will study the ankle dynamics by using reciprocal screws to the axis location in a re-configurable platform. The robot will be lightweight because of the use of cable-driven actuators, inspired by antagonistic muscles that work with reciprocal inhibition for energy optimization. The robot will reconfigure the structure, considering the ankle joint as a central mast, and referenced it with MMP and MLP markers.

Figure 39 shows a schematic of the re-configurable approach.

Other applications are, for example, by visualizing the platform trajectories one can explain how the calcaneal Achilles insertion is near to the platform’s A point. The platform’s normal vector changes abruptly near this region, as was depicted in Figure 24b and Figure 26a,b. Furthermore, Riemannian models have different properties. We will explore diagnosis and treatments based on the model and metrics by employing machine learning algorithms. This approach can be applied to other joints in humans and other animals, by designing specialized re-configurable hardware and software. Tracking the parameters in different ages and weight conditions, and comparing the ankle models in healthy and injured people.

## 5. Conclusions

Computer tomography (CT) and magnetic resonance (MR) images have greater precision and accuracy. Measurements in medical imaging will help us compare the errors (RMS) in the HAJ. In biomechanics, we have not found an ideal model for error comparison. Then, we will compare the error with an accurate measurement. The device has limitations regarding mechanical precision and deformation of its parts. We face up to the error through the electronic design system. The calibration process is imperative for enhancing accuracy.

The calibration process is human-dependent. We read the digital measurement and compare it with caliper measurements directly in the sensor. Then, we register the data in a table to find the equivalence. An electronic board with trimmers avoids saturation, bias, and calibration; a 10 bit ADC and an exponentially weighted moving average (EWMA) filter the noise signals. We have implemented a processing (Software) calibration interface. We avoid adding more specific technical data, such as CMRR, ADC speed, mechanical tolerance, and other issues inherent to the measuring devices.

Digital sensors, communications, and POE function fitting use machine learning techniques.

The ankle is the most commonly injured joint of the lower limb, fundamental to the human body’s balance; it is necessary to measure the range of motion by in vivo methods for patients in lying positions in reduced or remote places. The device’s development considers ankle anatomy and anthropometry. We propose a Riemannian manifold model based on the device’s data readings. Performing simulations enabled us to design the size of the device and the maximal length of the wires. We present a trilateration algorithm, projecting the tetrahedron’s sides on the base plane. The sensors are modular and part of the device’s lightweight and portable structure. The electronic system is modular, replaced by other single-board computers (SBC) and microcontroller unities. We will also use the TM for ankle characterization and diagnosis for rehabilitation robotics, prosthesis, and orthosis design. The prototype is not a finished product (the TRL is 2). The work’s scope is to validate the use of a modern alternative biomechanic representation of the human ankle joint. It is a platform for testing an alternative trilateration method that employs draw-wire sensors (DWS). Such sensors have a constant tension, coiled on a drum attached to a potentiometer, and a flat spiral spring. We also attempted to develop a flexible device design for several foot sizes. We are working on a newer device version with an enhanced attachment system, a more compact design, and digital DWS compatible with a configurable robot. Machine learning and edge computing will assist in disease diagnosis and rehabilitation of patients.

## Figures and Tables

**Figure 1 bioengineering-09-00199-f001:**
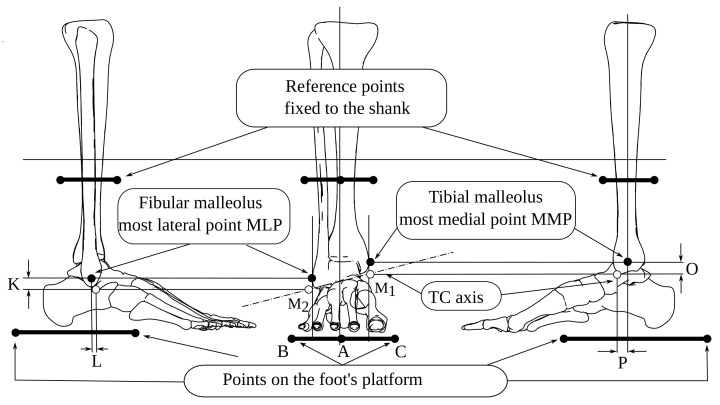
Reference points from anthropometric values K, L, O, and P.

**Figure 2 bioengineering-09-00199-f002:**
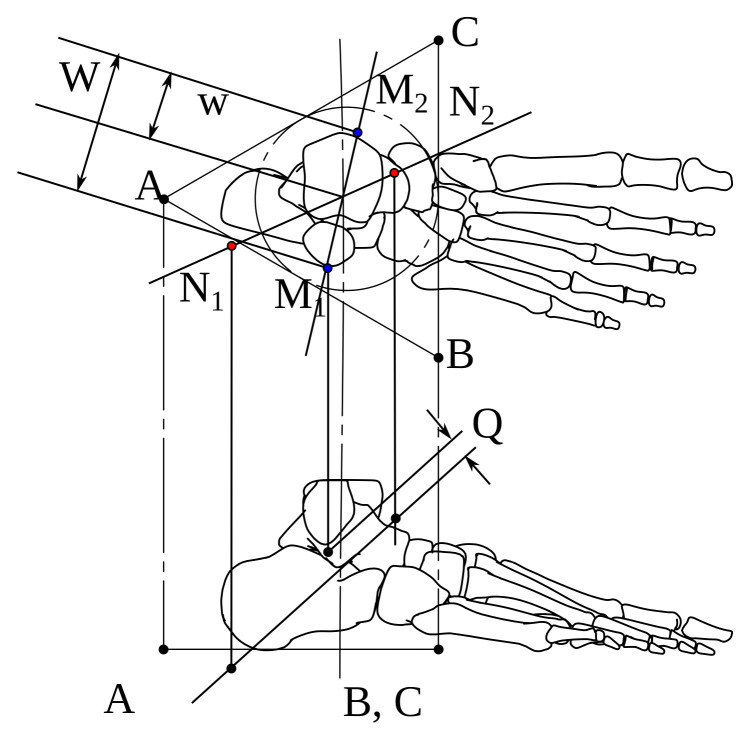
Q, W, and w distances from lateral and transverse views.

**Figure 3 bioengineering-09-00199-f003:**
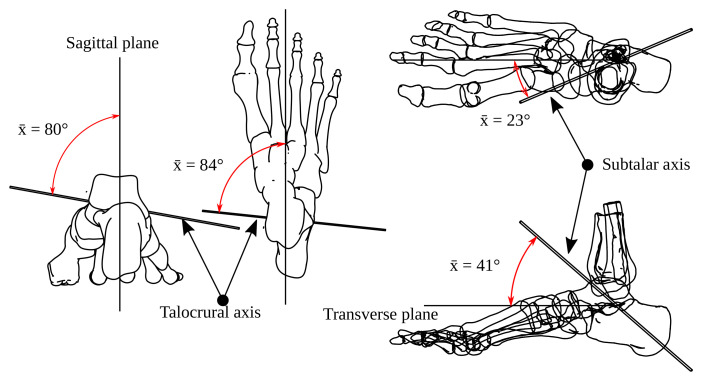
Mean relative position of the ST and TC axis.

**Figure 4 bioengineering-09-00199-f004:**
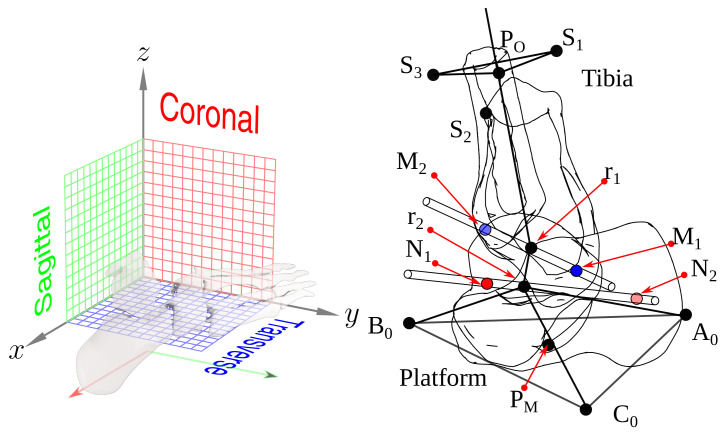
Planes, axes, and points of corresponding references.

**Figure 5 bioengineering-09-00199-f005:**
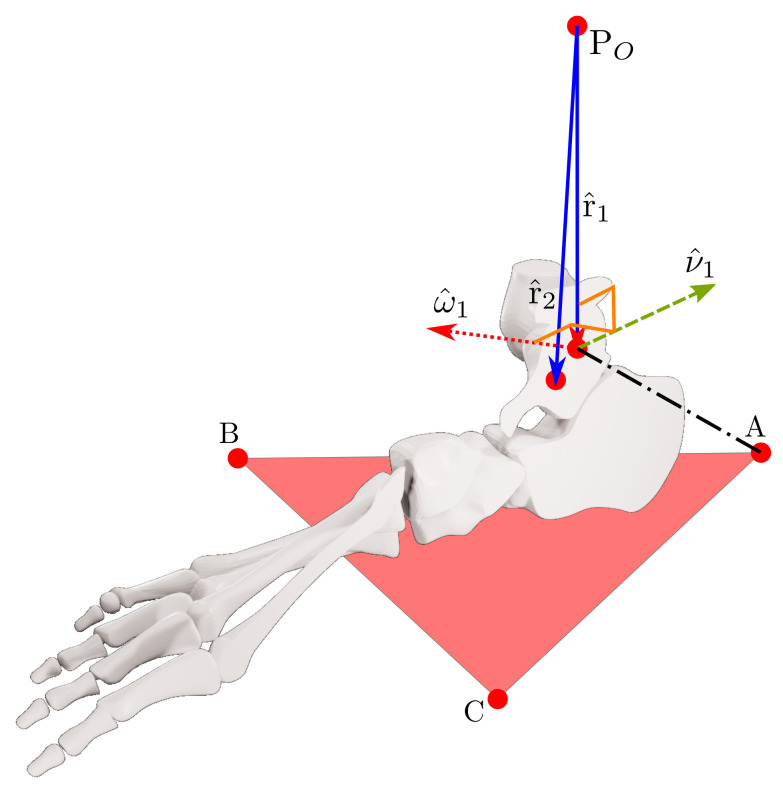
Vectors and points on the sagittal plane.

**Figure 6 bioengineering-09-00199-f006:**
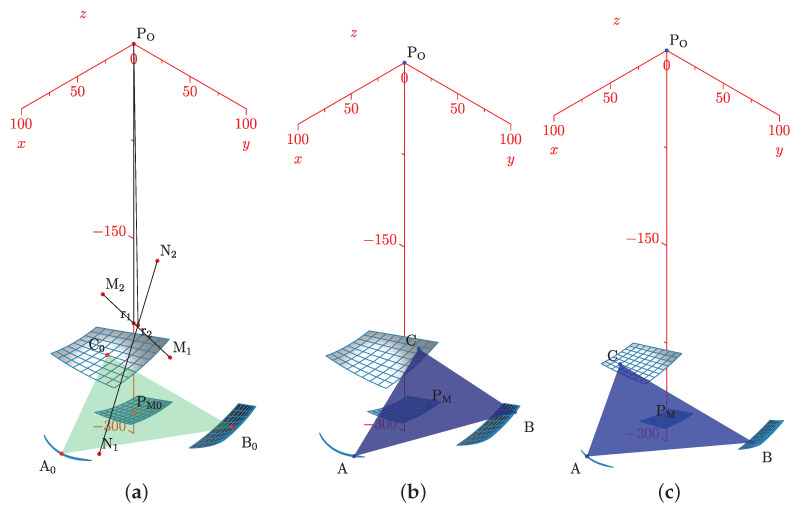
Forward kinematics for (**a**) initial position and (**b**) $\theta_{1range}=\theta_{2range}=\left[-15^{\circ },15^{\circ }\right],\theta_{1}=\theta_{2}=10^{\circ }$ and (**c**) $\theta_{1range}=\theta_{2range}=\left[-10^{\circ },10^{\circ }\right],\theta_{1}=\theta_{2}=5^{\circ }$.

**Figure 7 bioengineering-09-00199-f007:**
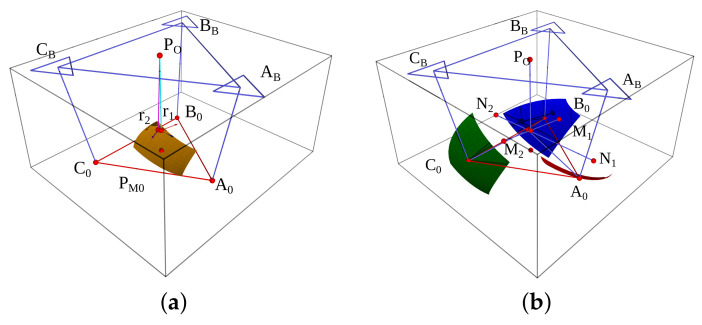
Geometric design: (**a**) is the platform center, base, and $r_{1},r_{2}$; and (**b**) platform vertices with talocrural and subtalar axis.

**Figure 8 bioengineering-09-00199-f008:**
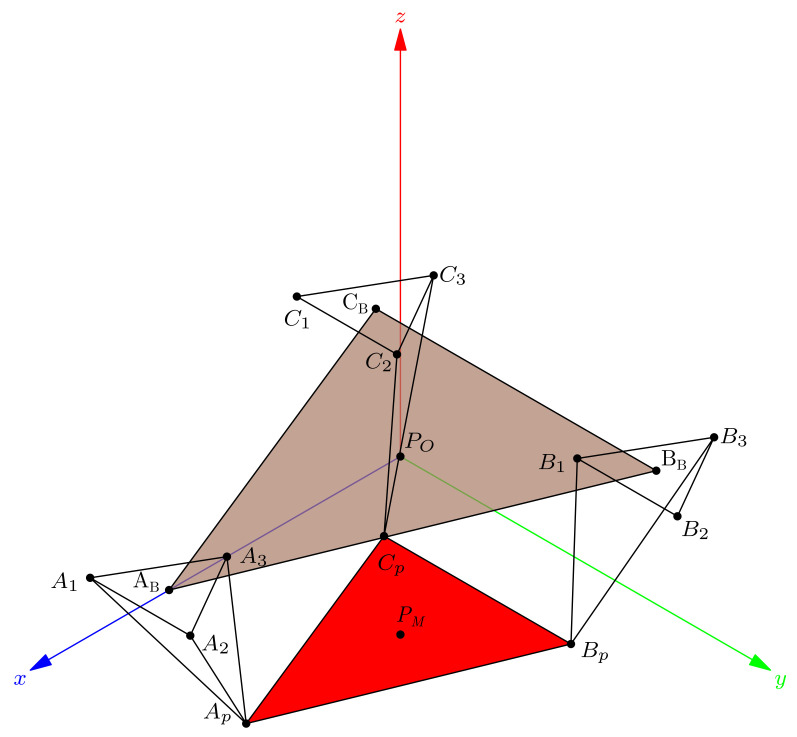
Geometric design of the DWS arrays.

**Figure 9 bioengineering-09-00199-f009:**
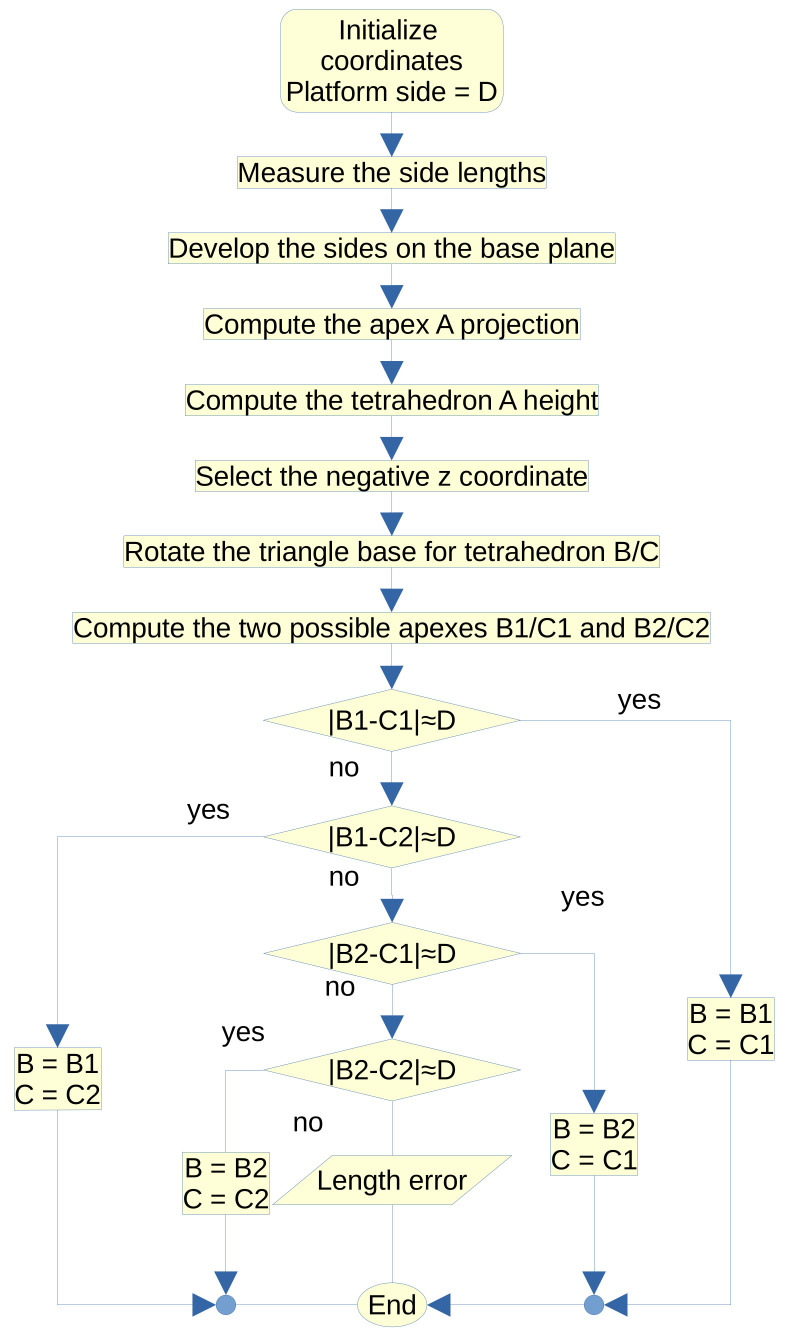
Tetrahedron trilateration flowchart.

**Figure 10 bioengineering-09-00199-f010:**
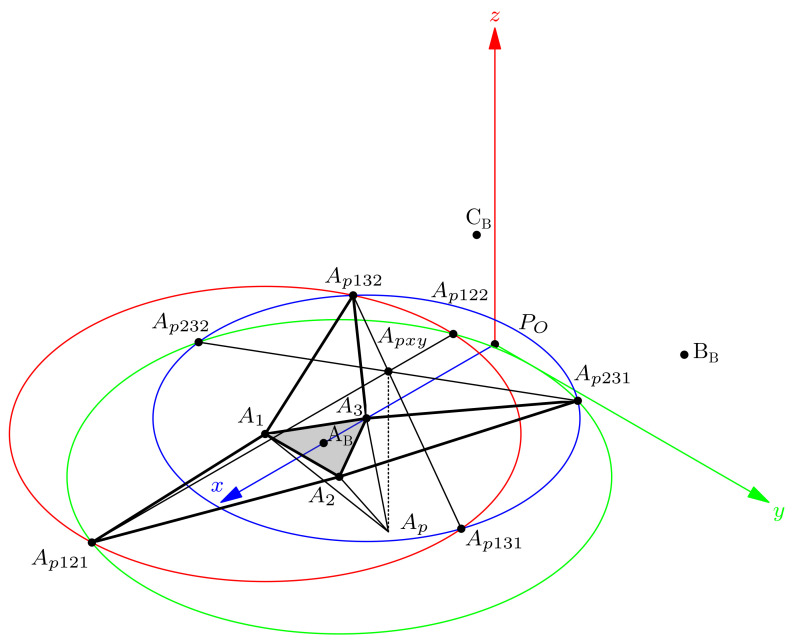
Finding the apex $A_{p}$.

**Figure 11 bioengineering-09-00199-f011:**
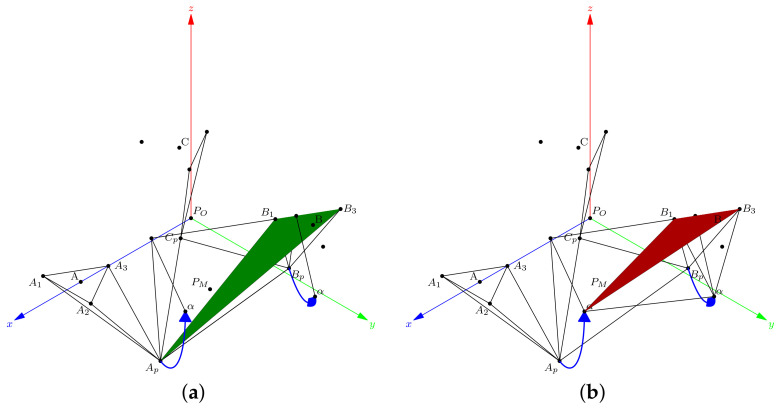
Rotation of α angle about the axis $B_{1}-B_{3}$: (**a**) original tetrahedron, $T_{B}$ (**b**) rotated tetrahedron.

**Figure 12 bioengineering-09-00199-f012:**
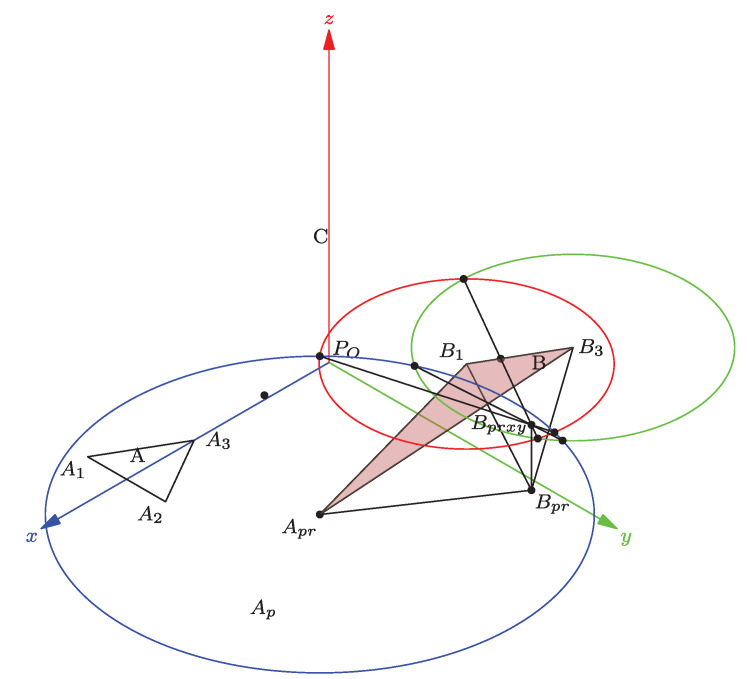
Finding the apex $B_{\mathrm{pr}}$.

**Figure 13 bioengineering-09-00199-f013:**
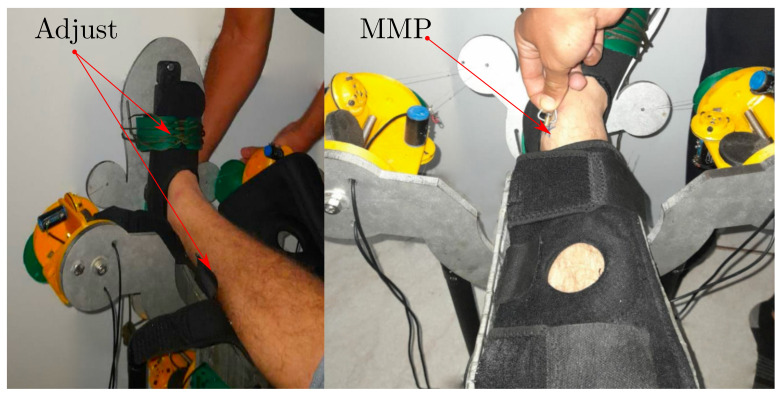
Adjusting the foot, the shank and the Most Medial Point reference.

**Figure 14 bioengineering-09-00199-f014:**
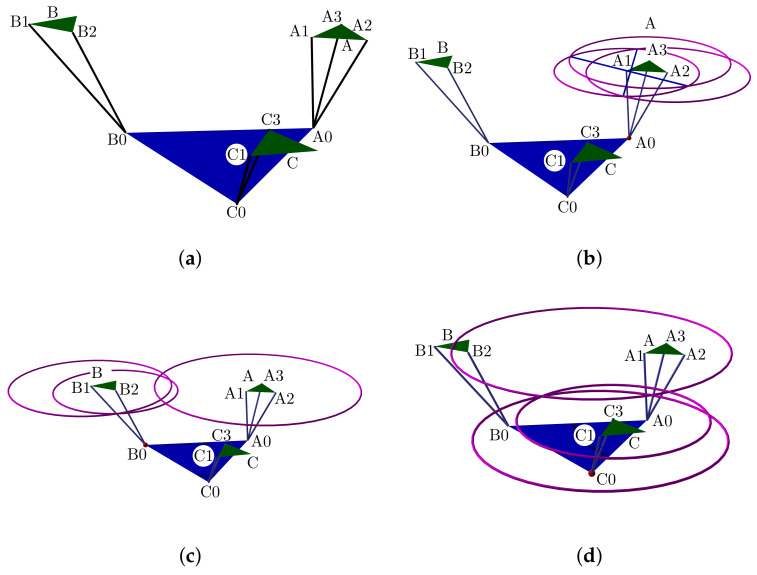
Computed positions from sensor lengths at portable configuration: (**a**) the rest position, (**b**) apex A, (**c**) apex B, and (**d**) apex C.

**Figure 15 bioengineering-09-00199-f015:**
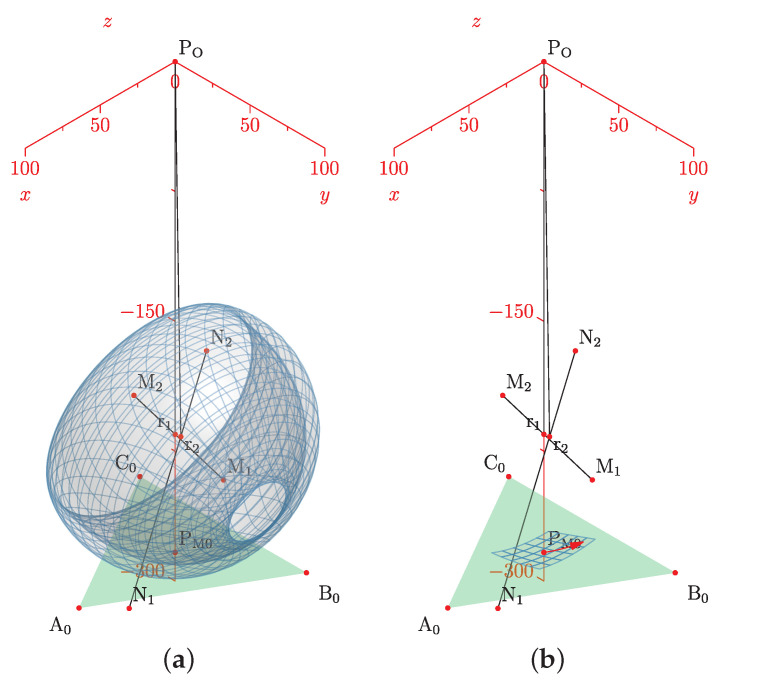
Simulation of the platform central point with variations in the mean statistical values: (**a**) platform’s center point manifold, (**b**) manifold chart and a geodesic.

**Figure 16 bioengineering-09-00199-f016:**
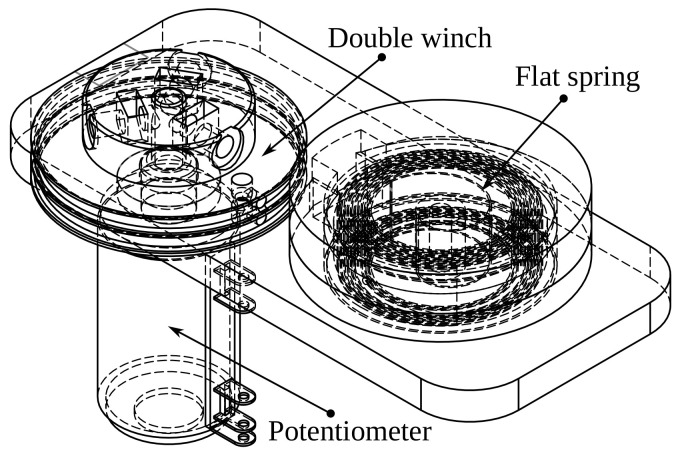
Draw-wire sensor design.

**Figure 17 bioengineering-09-00199-f017:**
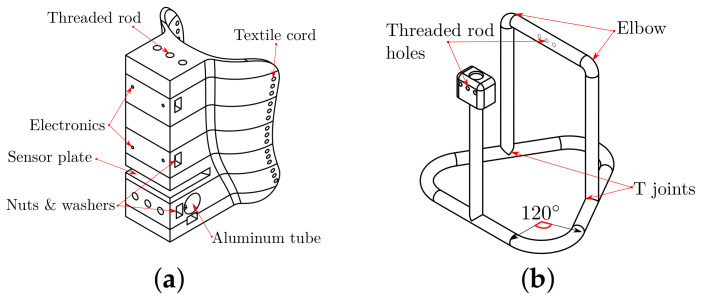
Mechanical attachment: (**a**) calf support and (**b**) aluminum tube structure.

**Figure 18 bioengineering-09-00199-f018:**
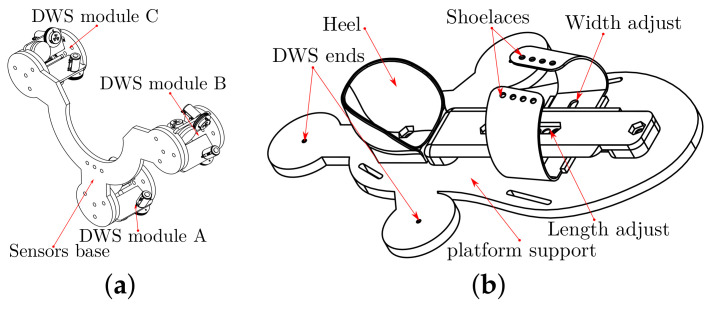
Base and platform: (**a**) DWS modules support (**b**) platform with foot’s size adjustment.

**Figure 19 bioengineering-09-00199-f019:**
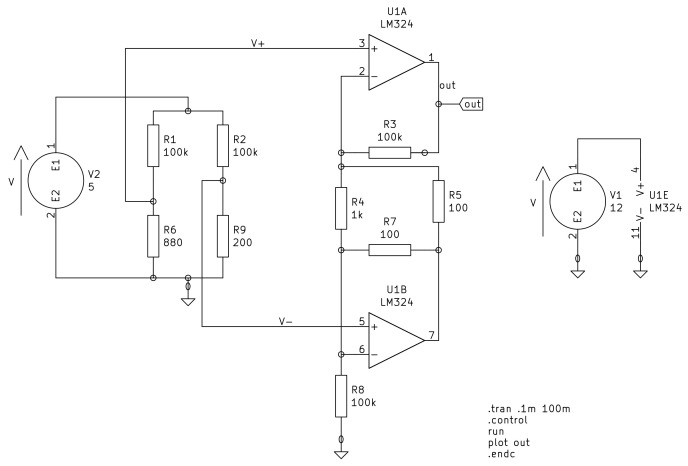
Two Op. Amp. instrumentation amplifier.

**Figure 20 bioengineering-09-00199-f020:**
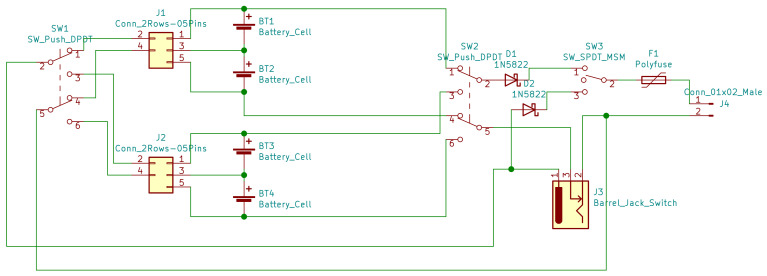
Power system with backup, BMS, boost, and buck converters.

**Figure 21 bioengineering-09-00199-f021:**
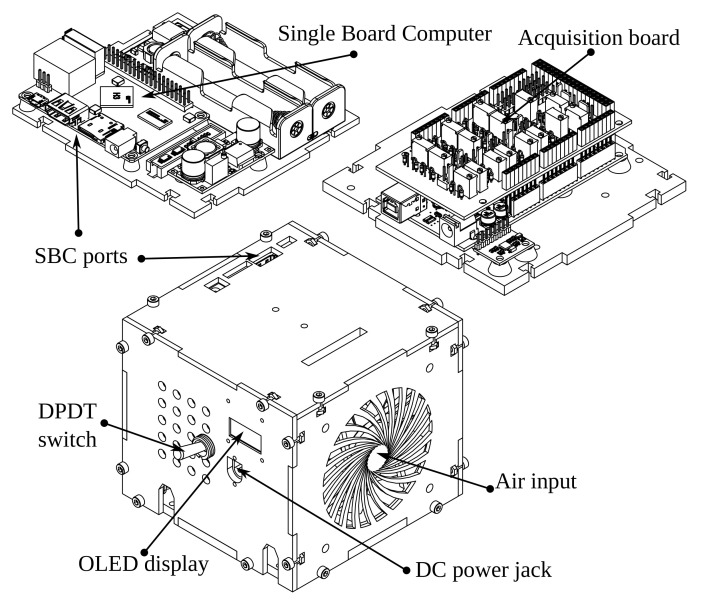
Modular electronics casing.

**Figure 22 bioengineering-09-00199-f022:**
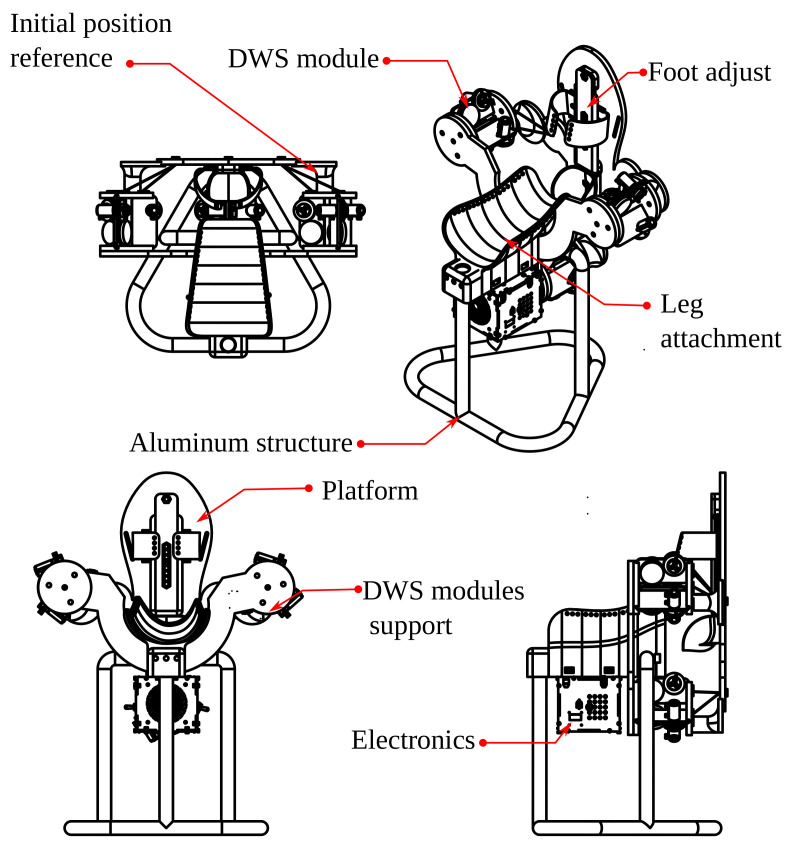
Complete prototype.

**Figure 23 bioengineering-09-00199-f023:**
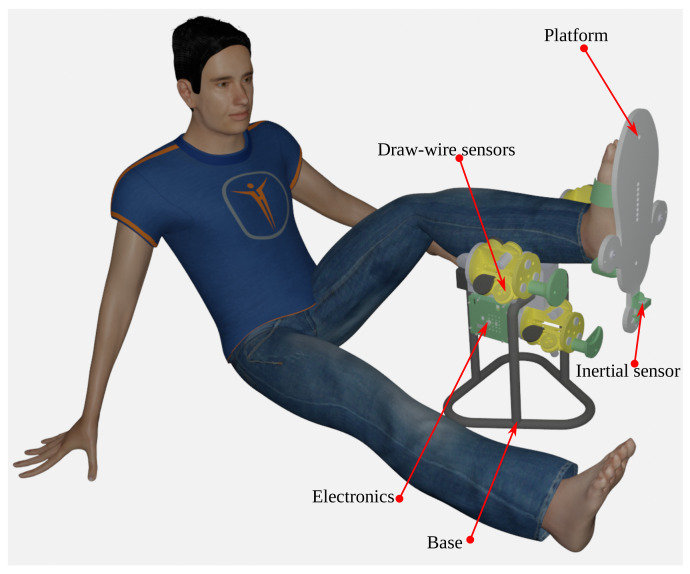
Rendered image with a 175 cm height patient.

**Figure 24 bioengineering-09-00199-f024:**
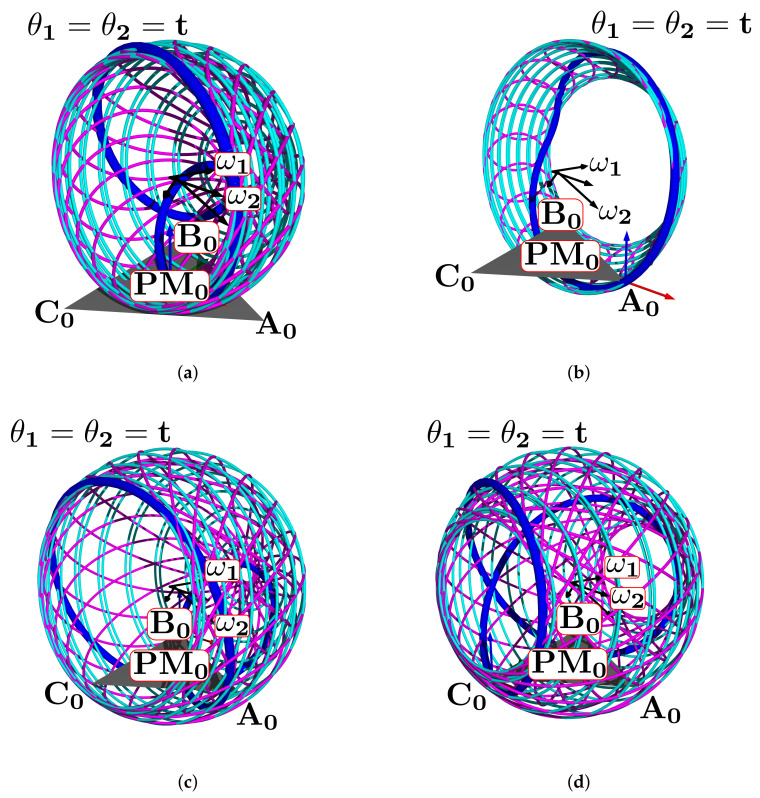
Simulation of all points: (**a**) platform’s central point, (**b**) attachment a, (**c**) attachment b, and (**d**) attachment c.

**Figure 25 bioengineering-09-00199-f025:**
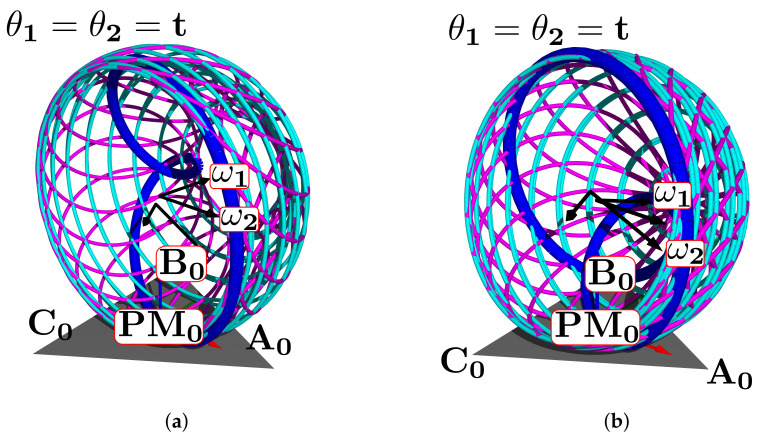
Simulation of the platform central point with variations in the mean statistical values: (**a**) 10% below, and (**b**) 10% over.

**Figure 26 bioengineering-09-00199-f026:**
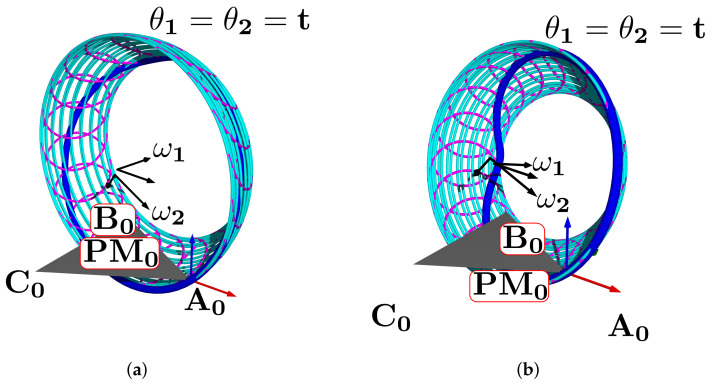
Simulation of the platform’s attaching point A: (**a**) mean values plus 10%, (**b**) mean values minus 10%.

**Figure 27 bioengineering-09-00199-f027:**
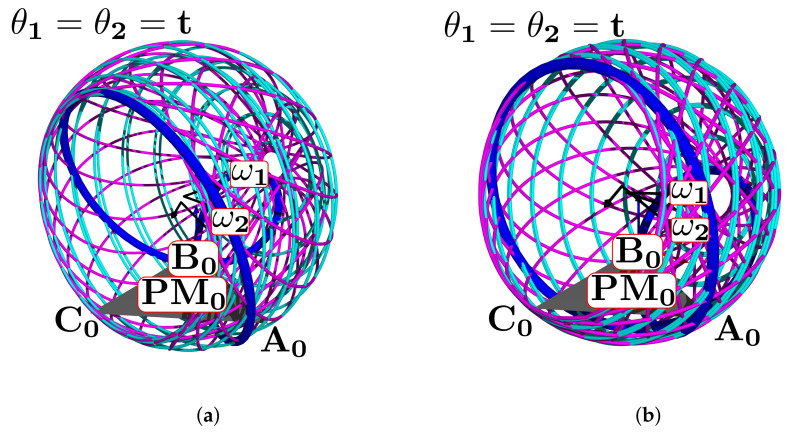
Attaching point B simulation: (**a**) adding 10% to the statistic mean values, (**b**) subtracting 10%.

**Figure 28 bioengineering-09-00199-f028:**
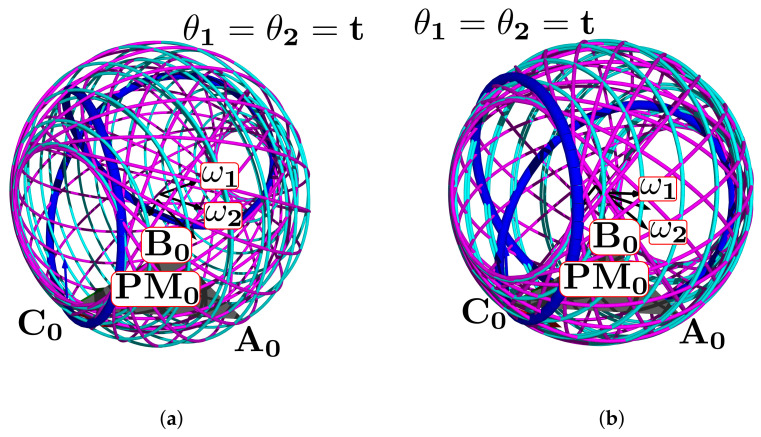
Simulation results for C: (**a**) mean values plus 10%, (**b**) mean values minus 10%.

**Figure 29 bioengineering-09-00199-f029:**
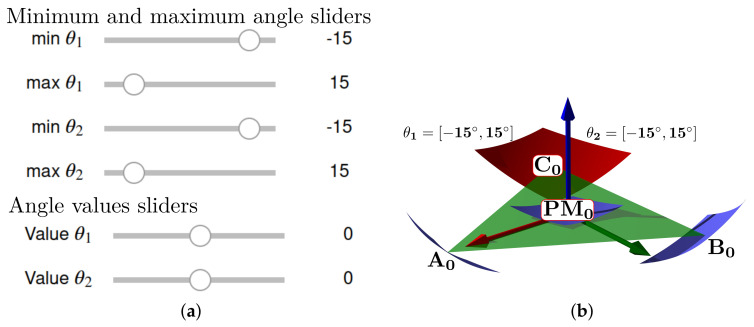
Interactive simulation example: (**a**) sliders, (**b**) rendering.

**Figure 30 bioengineering-09-00199-f030:**
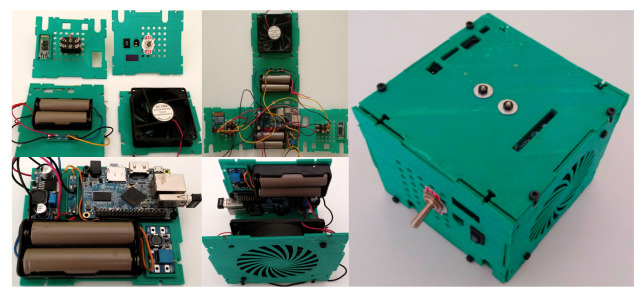
Connections and electronics.

**Figure 31 bioengineering-09-00199-f031:**
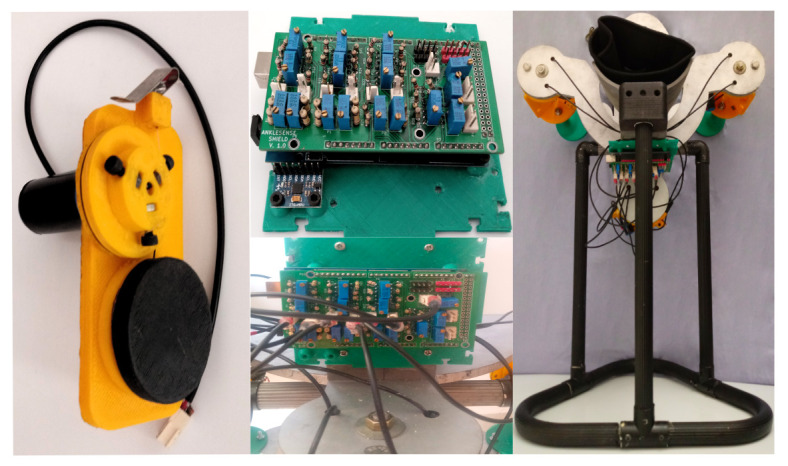
Assembled structure.

**Figure 32 bioengineering-09-00199-f032:**
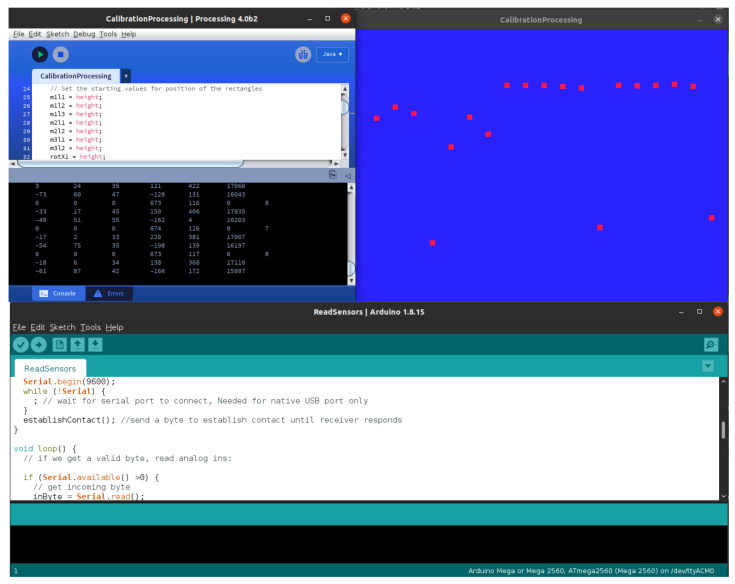
Processing calibration interface.

**Figure 33 bioengineering-09-00199-f033:**
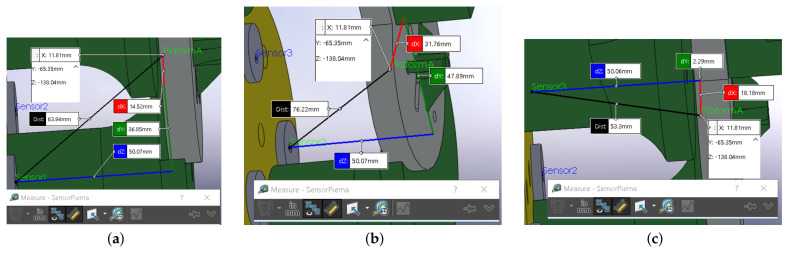
Measuring in SolidWorks (2017–2018 Student Edition, Dassault Systèmes, Vélizy-Villacoublay, France)^®^: (**a**) sensor 1, (**b**) sensor 2, (**c**) sensor 3.

**Figure 34 bioengineering-09-00199-f034:**
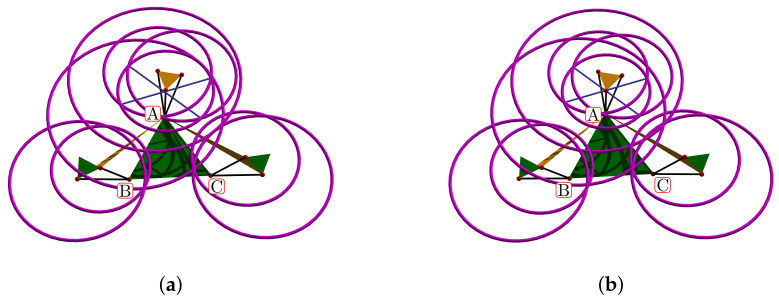
First two trilateration results: (**a**) position 1, (**b**) position 2.

**Figure 35 bioengineering-09-00199-f035:**
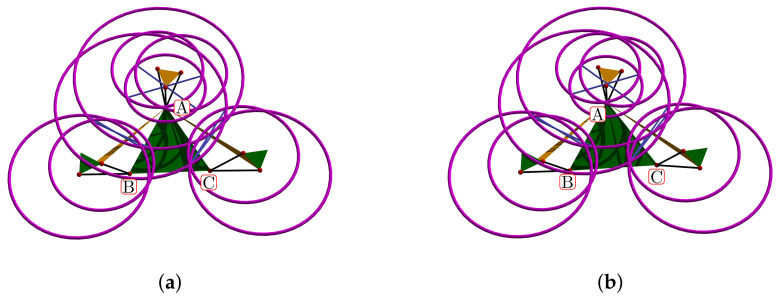
Latest two trilateration results: (**a**) position 3, (**b**) position 4.

**Figure 36 bioengineering-09-00199-f036:**
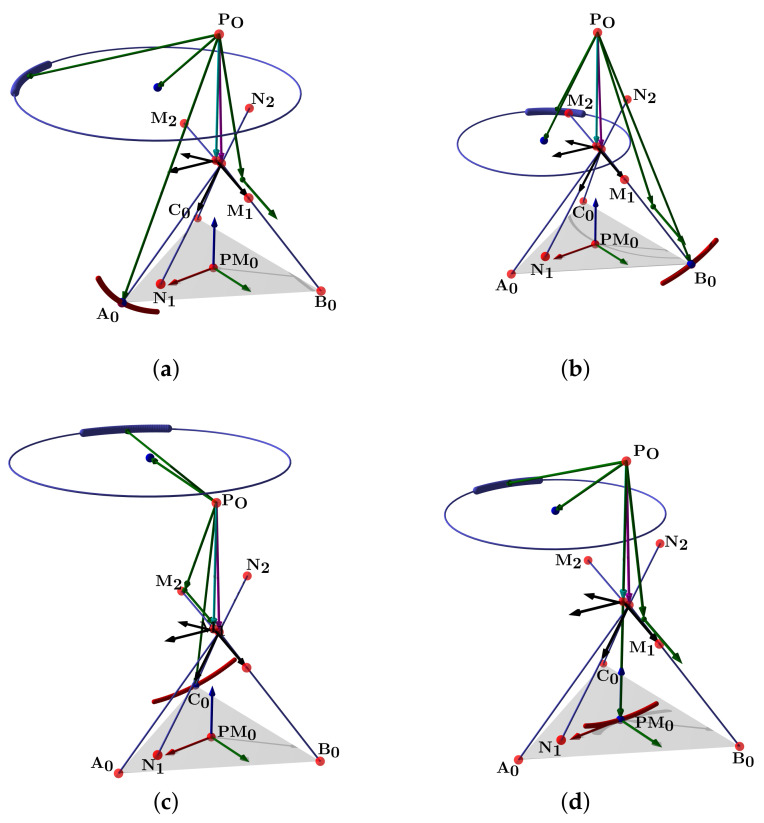
TC axis circle fitting: (**a**) trajectory A, (**b**) trajectory B, (**c**) trajectory C, (**d**) trajectory PM.

**Figure 37 bioengineering-09-00199-f037:**
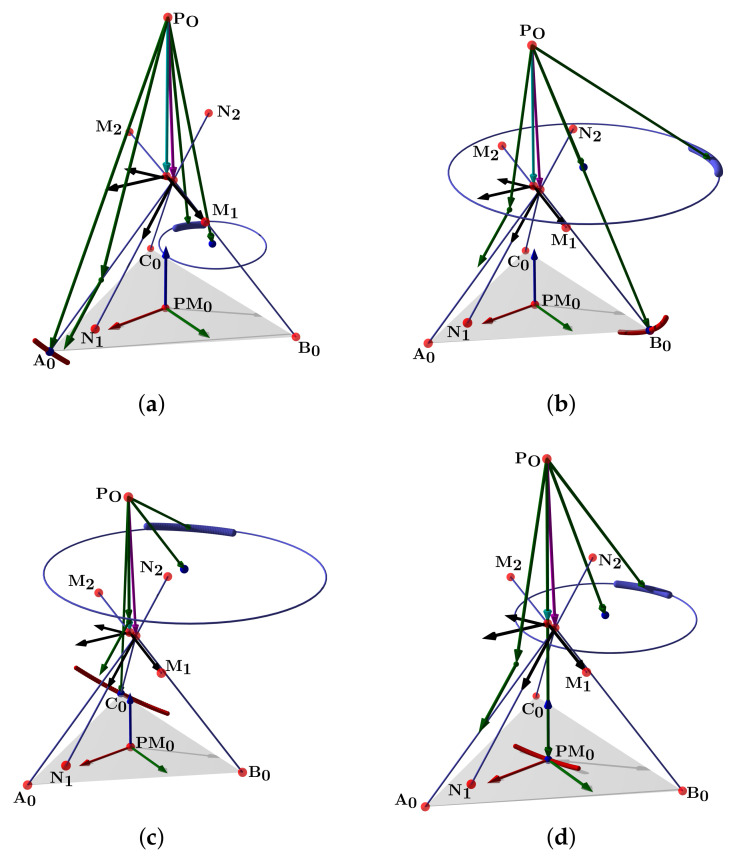
ST axis circle fitting: (**a**) trajectory A, (**b**) trajectory B, (**c**) trajectory C, (**d**) trajectory PM.

**Figure 38 bioengineering-09-00199-f038:**
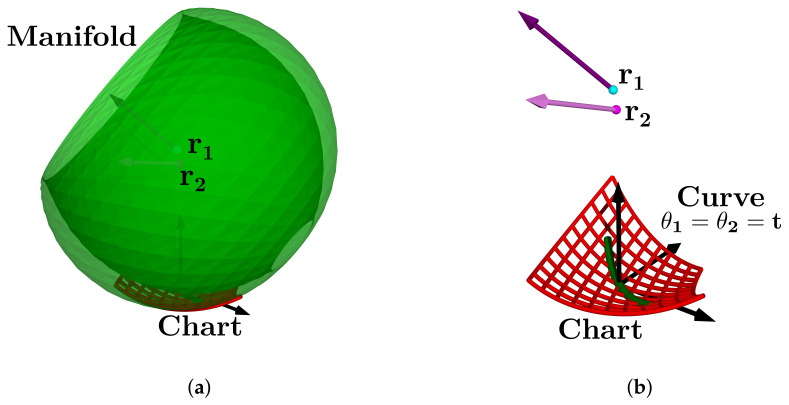
Ankle joint manifold. (**a**) Manifold for PM, (**b**) chart with ankle axis coordinates.

**Figure 39 bioengineering-09-00199-f039:**
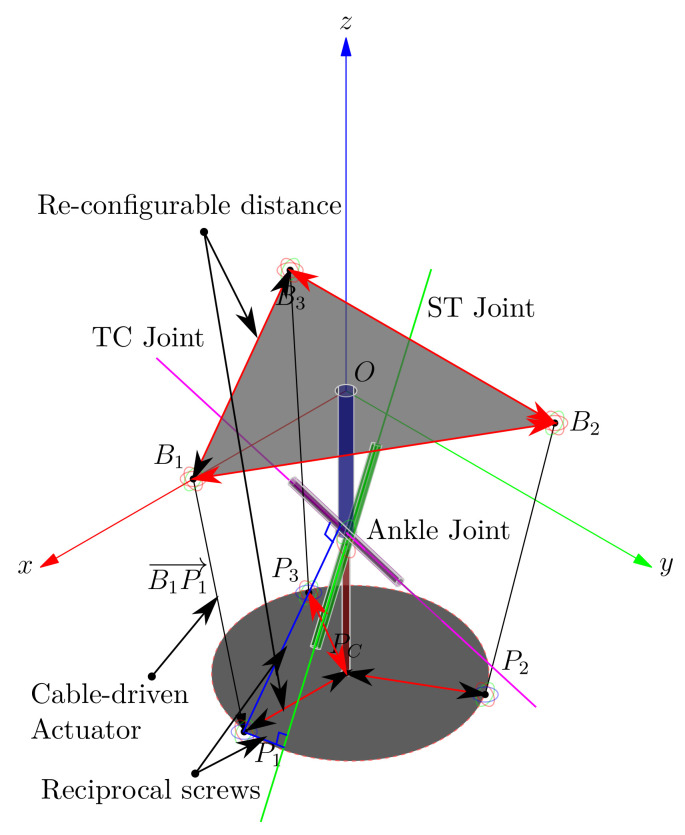
Re-configurable cable-driven robot concept.

**Table 1 bioengineering-09-00199-t001:** Mean values of anthropometric measurements.

Variable	K (cm)	L (cm)	O (cm)	P (cm)	Q (cm)	R = W/w
Mean	1.2 cm	1.1 cm	1.6 cm	1.0 cm	0.5 cm	0.54 cm

**Table 2 bioengineering-09-00199-t002:** Calibration results with digital measurements and real measurements.

Measurements	l1M1	l2M1	l3M1	l1M2	l2M2	l1M3	l2M3
BCD value	239	330	246	265	177	252	242
Vernier Caliper, cm	8.0 cm	5.3 cm	6.9 cm	13.0 cm	8.4 cm	7.8 cm	11.5 cm

**Table 3 bioengineering-09-00199-t003:** Error compared with SolidWorks^®^ measurements.

Measurements	l1M1	l2M1	l3M1
Measured distance	7.622 cm	5.33 cm	6.384 cm
Error in cm	0.38 cm	−0.030 cm	0.52 cm

**Table 4 bioengineering-09-00199-t004:** Sensor measurements in four different positions.

Positions	l1M1	l2M1	l3M1	l1M2	l2M2	l1M3	l2M3
Pos1., cm	11.0 cm	12.6 cm	12.5 cm	14.8 cm	10.8 cm	15.2 cm	11.9 cm
Pos2., cm	10.2 cm	11.7 cm	11.6 cm	15.2 cm	11.3 cm	15.5 cm	12.2 cm
Pos3., cm	9.40 cm	10.8 cm	10.8 cm	15.6 cm	11.7 cm	15.8 cm	12.5 cm
Pos4., cm	8.56 cm	9.89 cm	9.95 cm	16.0 cm	12.2 cm	16.0 cm	12.7 cm

**Table 5 bioengineering-09-00199-t005:** A, B and C coordinates computed from the four positions.

Positions	A	B	C
Pos1., cm	(−11.7, −1.06, −11.0) cm	(6.11, −9.77, −8.76) cm	(5.54, 8.81, −9.35) cm
Pos2., cm	(−12.1, −0.93, −10.2) cm	(5.62, −9.92, −9.37) cm	(4.83, 9.46, −10.1) cm
Pos3., cm	(−12.4, −0.65, −9.39) cm	(5.27, −9.79, −9.68) cm	(4.94, 9.03, −10.2) cm
Pos4., cm	(−12.7, −0.48, −8.53) cm	(4.68, −10.0, −10.3) cm	(3.54, 10.7, −11.1) cm

**Table 6 bioengineering-09-00199-t006:** TC axis circle fitting.

Trajectory	Center	Direction	Radius
A	(0.08649, 2.138, −6.712) cm	(−0.089, −0.95, 0.31)	7.666
B	(0.5713, 5.531, −7.824) cm	(−0.089, −0.95, 0.31)	5.246 cm
C	(−0.2442, −2.669, −5.315) cm	(−0.089, −0.95, 0.31)	7.206 cm
PM	(0.1552, 1.642, −6.683) cm	(−0.089, −0.95, 0.31)	5.375 cm

**Table 7 bioengineering-09-00199-t007:** ST axis circle fitting.

Trajectory	Center	Direction	Radius
A	(4.444, 1.825, −9.008) cm	(−0.75, −0.28, 0.60)	2.428 cm
B	(1.757, 0.6768, −6.925) cm	(−0.75, −0.28, 0.60)	6.567 cm
C	(0.1578, 0.1819, −5.807) cm	((−0.75, −0.28, 0.60)	6.935 cm
PM	(2.087, 0.8882, −7.281) cm	(−0.75, −0.28, 0.60)	3.875

**Table 8 bioengineering-09-00199-t008:** Axis estimation data.

Axis	Median Center	Median Normal	*r*	ω
TC	(1.92, 0.783, −7.10) cm	(−0.750, −0.280, 0.600)	(−0.174, 0.000, −5.43) cm	(−0.750, −0.280, 0.600)
ST	(0.121, 1.89, −6.70) cm	(−0.0890, −0.950, 0.310)	(−0.0562, 0.000, −6.08) cm	(−0.0890, −0.950, 0.310)

**Table 9 bioengineering-09-00199-t009:** Plucker line coordinates.

Axis	Plucker Line Coordinates
TC	[−0.750: −0.280: 0.600: −1.52: 4.17: 0.0487]
ST	[−0.0890: −0.950: 0.310: −5.78: 0.559: 0.0534]

## Data Availability

The code and CAD electronics and mechanical designs are available.

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
