# Peer review of "Turmell-Meter: A Device for Estimating the Subtalar and Talocrural Axes of the Human Ankle Joint by Applying the Product of Exponentials Formula"

_bioengineering, 2022, doi:10.3390/bioengineering9050199_

Round 1

Reviewer 1 Report

The article is interesting and may undoubtedly attract the attention of readers of this special issue. I only would to note that the kinematics is somewhat incomplete when infinitesimal kinematics is not considered. For example with it one could address the subject of singularities. On the other hand, the article is extremely long and could divert the readers' attention. For example, Im not sure if it is really necessary to include Appendix B. Otherwise the content and treatment of the article is such that it deserves to be considered for publication.

Author Response

Thank you very much for your acceptance.
    • Regarding the infinitesimal kinematics, we will address the singularities issue in a future assistive robot. In order to avoid extending the current article, we do not include it. In the Turmell-Meter design, we only considered the axes representation and estimation, and focused on the human motion-based model, also in the hardware and software implementation. We will refine the model with functional approximation and model optimization using machine learning techniques. 

    • We will suppress the appendixes and put in the references the address to GitHub sources in an interactive notebook for web visualization. 

Reviewer 2 Report

The authors present a good introduction to their work but do not present the plan of the publication or its objectives. 

The beginning of the article presents anthropomorphic data. This part of the article is well presented. However, I don't know why it is the average man that is studied. Can we make a mechanism that fits 90% of the percentiles? in size and range of motion? The foot movements are well defined. Only the axes of rotation of the angles \theta_1 and \theta_2 are to be specified. Two articles cited are with ?? Units should be placed in the angles at all times.

The choice of a 7-cable system must be justified. Why not 9 cables? or less than 7? The design approach and choice of cable placement are a bit confusing. How to ensure that there are no singularities? Can there be a unique solution for the inverse geometric model? Are the cables always tensioned?  I think that the motion theory should be grouped together and the mechatronic system presented afterward. I do not understand section 2.5.2. Please clarify why there is equation redundancy. Can we simplify the problem?

The prototype seems to be of good quality and functional.
Attention, some letters are written with latex errors: theta_2.

Author Response

  • We added a plan and objectives to the introduction.

  • We included the limitations in size and range of motion based on official statistics, also we only mention the anthropometry of most cited studies and remove the average body height. In addition, we excluded the cites that correlate with the proportions of ankle and body height. We explained the model simplification and corrected the citation and angle notation in the entire manuscript.

  • We explain why we chose the seven-cable system, and justify why not study the singularities for measuring only purposes in a healthy average ankle joint, with no singularities along its range of motion. Also, we take care of the singularities and multiple solutions algorithmically, by ensuring the platform distances from a known starting position, extracted from the CAD model named as “rest position”. Likewise, we added a flowchart explaining the method. We reorganize the article, firstly presenting the motion study and then the mechatronic system. Finally, we made clarification and simplification in section 2.5.2.

  • Thank you for this appreciation. We made the LaTeX corrections.

Reviewer 3 Report

The topic is promising, however, the manuscript must be improved in the following aspects:

  1. The abstract section presents the problem, the method, but the results should show the relevant performance metrics, such as error, precision, or others. Likewise, in the conclusion section of the abstract, the scope and limitations, applications, and TRL of the device (Turmell-meter) must be shown.
  2. Rewrite the introduction describing the context of troublesome to take in vivo measurements of the ankle joint, state of the art, to present the Turmell-meter highlighting the contribution compared with the state of the art, and describe in brief the sections of comprising the manuscript.
  3. The abbreviation should follow the next form:  Human Ankle Joint (HAJ), Computer-Aided Design (CAD)...
  4. SolidWorks®
  5. Digital Measurements instead of digital lectures
  6. In tables shows the units 
  7. To show figure 29 complete (missing an edge of sides a and b)
  8. The conclusions section should describe in detail the error (RMS) or the accuracy, the scope and limitations, applications, TRL of the device (Turmell-meter), and future works.

Author Response

The conclusions now include Technology Readiness Level 2 (TRL 2: Applied Research: Initial practical applications are identified. Potential of material or processes to solve a problem, satisfy a need, or find an application is confirmed).

We explain the limitations and accuracy issues. Also, we mentioned how we face up to the error through the electronic design system. The calibration process is human dependent. We read the digital measurement and compare with caliper measurements in the sensor. Then, we register the data in a table to find the equivalence. We designed an electronic board with trimmers for rough calibration, to avoid saturation and to reduce the bias.  We used a 10 bit ADC, and an Exponentially Weighted Moving Average (EWMA). Also, we implement a Processing (Software).   We avoid adding more specific technical data such as CMRR, ADC speed, mechanical tolerance and other issues inherent to the measuring devices.

We added future work, enhanced by digital sensors, communications, and function fitting through machine learning techniques.